# Tunable Domain Adaptation Using Unfolding

## Abstract

Machine learning models often struggle to generalize across domains with varying data distributions, such as differing noise levels, leading to degraded performance. Traditional strategies like personalized training, which trains separate models per domain, and joint training, which uses a single model for all domains, have significant limitations in flexibility and effectiveness. To address this, we propose two novel domain adaptation methods for regression tasks based on interpretable unrolled networks—deep architectures inspired by iterative optimization algorithms. These models leverage the functional dependence of select tunable parameters on domain variables, enabling controlled adaptation during inference. Our methods include Parametric Tunable-Domain Adaptation (P-TDA), which uses known domain parameters for dynamic tuning, and Data-Driven Tunable-Domain Adaptation (DD-TDA), which infers domain adaptation directly from input data. We validate our approach on compressed sensing problems involving noise-adaptive sparse signal recovery, domain-adaptive gain calibration, and domain-adaptive phase retrieval, demonstrating improved or comparable performance to domain-specific models while surpassing joint training baselines. This work highlights the potential of unrolled networks for effective, interpretable domain adaptation in regression settings.

## 1 Introduction

Machine learning models typically learn input-output mappings by optimizing parameters to minimize prediction errors on paired data. Their ability to generalize effectively to unseen data hinges on the assumption that both training and test data are drawn from similar distributions. However, this assumption often breaks down when data varies due to domain-specific factors, such as noise levels, leading to performance degradation. To ensure robust model performance in such settings, it becomes critical to account for this variability during training.

A straightforward strategy to handle domain shifts is *Personalized Training (PT)*, where separate models are trained for each domain. While this yields strong performance, it becomes impractical when the domain parameters vary continuously, and it also requires precise domain information during inference. On the other hand, *Joint Training (JT)* consolidates data from all domains into a single model, eliminating the need for domain-specific information. However, JT typically underperforms compared to PT, as it attempts to learn a unified representation across varying domains, which may not capture domain-specific nuances effectively.

To address the issues arising in the PT and the JT approaches, *transfer learning* and *domain adaptation* techniques have been proposed, often outperforming both PT and JT. In classification tasks, these methods commonly involve retraining only the final layers of a model, under the assumption that the feature extractor is domain-invariant Yosinski et al. (2014); He et al. (2016); Devlin et al. (2019); Pan & Yang (2010). However, identifying which layers should serve as feature extractors is non-trivial Tseng et al. (2023). In He et al. (2024), the authors describe a gradual domain adaptation method using optimal transport to improve classification performance across large domain shifts. In Farahani et al. (2021), Farahani et al. describe domain adaptation methods that address domain shifts by aligning features between domains, learning deep domain-invariant representations, iteratively refining models with pseudo-labels on target data, and using adversarial training to make features indistinguishable across domains. These require separate domain-specific data for progressive retraining during adaptation. But, all these approaches do not extend well to regression

tasks—which is the primary focus of this work, except for works in Singh & Chakraborty (2019); Obst et al. (2021); Li et al. (2022); Mao et al. (2023).

Our goal is to develop domain adaptation methods suitable for regression problems where the underlying assumption is that the domains are parametric. While Deep Neural Networks (DNN) and Convolutional Neural Networks (CNN)-based methods have demonstrated strong performance, their non-interpretable nature obscures the internal mechanisms of the network, making it difficult to determine which parameters should be adjusted for domain adaptation.

In this paper, we address these limitations by adopting *interpretable unrolled networks* Monga et al. (2021)— deep learning architectures derived by unrolling iterative optimization algorithms into finite-layer neural networks. These models combine the structure of classical optimization with the flexibility of deep learning, leading to improved interpretability. When trained on data with domain variability, the learnable parameters of unrolled networks often exhibit functional dependence on the domain parameter. Importantly, only a subset of these parameters—referred to as *tunable parameters*—tend to vary significantly with the domain, while the rest remain relatively invariant. Because the network is based on known algorithms, it allows us to uncover the functional relationship between the domain and its corresponding parameters. This allows us to model tunable parameters as functions of the domain parameter, enabling controlled adaptation during inference.

With an objective of designing a single network that effectively adapts to multiple domains, we propose two novel domain adaptation methods:

- *Parametric Tunable-Domain Adaptation (P-TDA):* This method assumes that domain parameters are either known for each data instance or that there exists a predefined function linking domain parameters to the tunable parameters of the unrolled network. The model is trained to minimize prediction error across domains by adjusting its parameters based on domain-specific characteristics. During inference, the model dynamically adapts by tuning the relevant parameters according to the known domain value.

- *Data-Driven Tunable-Domain Adaptation (DD-TDA):* In cases where neither the domain parameters nor the parametric relationship is known, this method learns a function that maps input data to domain-specific tunable parameters. During inference, the model simultaneously predicts outputs and adapts its parameters in a self-supervised fashion, using only the input data.

We demonstrate the effectiveness of these methods through two applications in *Compressed Sensing (CS)* Donoho (2006), a domain where the objective is to recover sparse signals from a limited number of measurements. The Iterative Shrinkage-Thresholding Algorithm (ISTA) Daubechies et al. (2004) is a commonly used, interpretable, mathematically derived method for solving the CS problem that can be naturally converted into an unrolled network. Building on this, unrolled optimization methods like Learned-ISTA (LISTA) Gregor & LeCun (2010)—a learned, unrolled variant of ISTA—have gained popularity for their balance of speed and interpretability, modeling each ISTA iteration as a separate layer in the network.

- *Noise-Adaptive LISTA:* In our first application, we focus on sparse signal recovery under varying noise conditions. Prior work Donoho & Johnstone (1994) has shown a strong relationship between the noise standard deviation and the soft-thresholding parameter in LISTA. Here, the *domain parameter* is the noise standard deviation (assuming Gaussian noise), and the *tunable parameter* is the soft-threshold parameter. Both P-TDA and DD-TDA are applied in this context. P-TDA leverages known noise levels to train a black-box model for dynamically adjusting the soft-thresholding parameter. In contrast, DD-TDA infers the relationship directly from data, learning how to adapt without access to the noise level during inference. Several experiments revealed that our methods perform at least as well as PT—and often better—in terms of the defined hit rate metrics, while also achieving lower estimation error compared to the JT method.

- *Domain-Adaptive Gain Calibration:* In our second application, we extend the proposed framework to a different compressed sensing problem involving multiplicative distortions modeled via unknown

sensor gains. Here, the domain variability arises due to the changes in sensor gains. Using explicit gain information (P-TDA) or data directly (DD-TDA), we demonstrate that both P-TDA and DD-TDA can efficiently adjust to such gain-induced variability by learning gain-compensated network parameters. Our results demonstrated that the suggested adjustable models beat joint training baselines in both structured and random gain cases, while achieving performance competitive with domain-specific models.

- *Domain-Adaptive Phase Retrieval:* As a third application, we investigate the non-linear problem of sparse phase retrieval, where only the magnitudes of linear measurements are available. Unlike the previous cases, the key domain parameter here is the signal sparsity level, which directly influences recovery performance but is often unknown. In the P-TDA setting, we assume access to the true sparsity and adapt the network accordingly. In the DD-TDA framework, we develop a lightweight sparsity estimation module that infers this parameter directly from the measurements. Our results show that both P-TDA and DD-TDA outperform PT and JT baselines, further demonstrating the flexibility of our approach in handling nonlinear inverse problems.

The paper is organized as follows. In Section 2, we formulate the general learning problem in a domain-parameterized setting. In Section 3, we present the proposed domain-adaptation method through unrolling-based networks. Three applications of the proposed approach for noise adaptation, gain calibration, and phase retrieval are discussed in Sections 4, 5, and 6, followed by conclusions.

## 2  Problem Formulation

Consider a learning problem from a set of paired data points $\mathcal{D} = \{\mathbf{y}_n, \mathbf{x}_n\}_{n=1}^N$ where $\mathbf{y}_n \in \mathbb{C}^{N_y}$ is input or feature to a network to be learned and $\mathbf{x}_n \in \mathbb{C}^{N_x}$ is the corresponding output or label. The terms $N_y$ and $N_x$ denote the dimensions of the input and output vectors, respectively. Here, $N$ represents the sample count of the dataset. The objective is to learn a function or a network $f_{\boldsymbol{\alpha}, \boldsymbol{\beta}} : \mathbb{C}^{N_y} \to \mathbb{C}^{N_x}$, with the learnable parameters $(\boldsymbol{\alpha}, \boldsymbol{\beta})$, such that the estimate $f_{\boldsymbol{\alpha}, \boldsymbol{\beta}}(\mathbf{y}_n)$ is close to the true output $\mathbf{x}_n$ in a predefined distance metric. The goal is typically achieved by fine-tuning the learnable parameters by solving the following optimization problem:

$$\min_{\boldsymbol{\alpha}, \boldsymbol{\beta}} \sum_{(\mathbf{y}_n, \mathbf{x}_n) \in \mathcal{D}} d\left(\mathbf{x}_n, f_{\boldsymbol{\alpha}, \boldsymbol{\beta}}(\mathbf{y}_n)\right), \tag{1}$$

where $d(\cdot, \cdot)$ is a distance metric in $\mathbb{C}^{N_x}$. The parameters learned through the process are optimal for the training data $\mathcal{D}$. It is expected that for any unseen data pairs, the learned network will provide fairly accurate results, given that the unseen or test data is similar to the training data. The similarity is generally ascertained by assuming that the train and test data are samples from the same probability distribution.

In many practical applications, the data set could be parameterized as $\mathcal{D}_{\boldsymbol{\theta}}$ where $\boldsymbol{\theta}$ is the parameter vector whose entries could be either discrete or continuous values. For example, $\boldsymbol{\theta}$ could represent the noise level in the features/input. Since the data changes with $\boldsymbol{\theta}$, an important question arises on how to train the network in this scenario. For the simplicity of the discussion, let us assume that $\boldsymbol{\theta}$ can take on one of the following $J$ values: $\{\boldsymbol{\theta}_j\}_{j=1}^J$. Then we can have the following training options.

The first is to train individual networks for each data set $\mathcal{D}_{\boldsymbol{\theta}_j}$. Such PT approach results in the learnable parameters $\{\boldsymbol{\alpha}_j, \boldsymbol{\beta}_j\}$ as a result of the following optimization problem,

$$\{\boldsymbol{\alpha}_j, \boldsymbol{\beta}_j\} = \arg\min_{\boldsymbol{\alpha}, \boldsymbol{\beta}} \sum_{(\mathbf{y}_n, \mathbf{x}_n) \in \mathcal{D}_{\boldsymbol{\theta}_j}} d\left(\mathbf{x}_n, f_{\boldsymbol{\alpha}, \boldsymbol{\beta}}(\mathbf{y}_n)\right). \tag{2}$$

The PT method requires training $J$ networks, which may not be practically feasible, especially when $\boldsymbol{\theta}$ is a continuous variable. In addition, during inference, the precise value of $\boldsymbol{\theta}$ should be known such that the corresponding network is used. Any mismatch may amplify the inference error.

To reduce the number of networks to be trained, a JT strategy could be used. Here, a single network is trained for all the data as

$$\{\boldsymbol{\alpha}_{\text{joint}}, \boldsymbol{\beta}_{\text{joint}}\} = \arg\min_{\boldsymbol{\alpha}, \boldsymbol{\beta}} \sum_{j=1}^{J} \sum_{(\mathbf{y}_n, \mathbf{x}_n) \in \mathcal{D}_{\boldsymbol{\theta}_j}} d\left(\mathbf{x}_n, f_{\boldsymbol{\alpha}, \boldsymbol{\beta}}(\mathbf{y}_n)\right). \tag{3}$$

Unlike PT, in the JT approach, knowledge of the precise value of $\boldsymbol{\theta}$ is not required. However, this approach may have an inferior performance compared with PT due to combined training. The reduction in performance depends on how the data parameter $\boldsymbol{\theta}$ is related to the data and hence, varies for different models.

An alternative and, perhaps, a more suitable solution to the problem is to apply methods from transfer learning and domain adaptation. In transfer learning, a general-purpose pre-trained network, let us say, on data set $\mathcal{D}_{\boldsymbol{\theta}_1}$, is retrained for a new data set, say $\mathcal{D}_{\boldsymbol{\theta}_2}$ Yosinski et al. (2014); He et al. (2016); Devlin et al. (2019); Pan & Yang (2010). While retraining, most layers are fixed, and the remaining ones, usually the last few, are relearned for the new data. For example, let $f_{\boldsymbol{\alpha}_1, \boldsymbol{\beta}_1}$ is the network learned on $\mathcal{D}_{\boldsymbol{\theta}_1}$. Then, during fine-tuning or retraining, the parameters $\boldsymbol{\alpha}_1$ are kept fixed, and the parameters $\boldsymbol{\beta}$ are learned. The pretraining and fine-tuning steps for the example considered here are given below.

$$\text{Pre-training: } \{\boldsymbol{\alpha}_1, \boldsymbol{\beta}_1\} = \arg\min_{\boldsymbol{\alpha}, \boldsymbol{\beta}} \sum_{(\mathbf{y}_n, \mathbf{x}_n) \in \mathcal{D}_{\boldsymbol{\theta}_1}} d\left(\mathbf{x}_n, f_{\boldsymbol{\alpha}, \boldsymbol{\beta}}(\mathbf{y}_n)\right).$$

$$\text{Fine-tuning: } \{\boldsymbol{\beta}_2\} = \arg\min_{\boldsymbol{\beta}} \sum_{(\mathbf{y}_n, \mathbf{x}_n) \in \mathcal{D}_{\boldsymbol{\theta}_2}} d\left(\mathbf{x}_n, f_{\boldsymbol{\alpha}_1, \boldsymbol{\beta}}(\mathbf{y}_n)\right).$$

The fine-tuning-based method to adapt a network for new data is typically applied in classification tasks where the first few layers are assumed to act as feature extractors. The last few layers, which are retrainable, work as classifiers. The features could be the same for different data sets; hence, only the classifier must be tuned. However, in conventional DNNs/CNNs, it is not straightforward to differentiate between feature extractor layers and classification layers. The method may also not be directly applicable to regression tasks, which are of interest in this work.

In the previous approach, once the network is pre-trained on data $\mathcal{D}_{\boldsymbol{\theta}_1}$, the data is no longer used during fine-tuning for dataset $\mathcal{D}_{\boldsymbol{\theta}_2}$. Note that the data from $\mathcal{D}_{\boldsymbol{\theta}_1}$ and $\mathcal{D}_{\boldsymbol{\theta}_2}$ have different distributions, and in the fine-tuning approach, there is no emphasis on distribution alignment. On the other hand, in the supervised domain-adaptation methods, data from all the distributions are used simultaneously to train a network, as in equation 3, with an additional term that explicitly reduces distribution differences between the data Long et al. (2013; 2015); Motiian et al. (2017). Though these methods seem suitable for the problem considered in this paper, the approaches do not consider the parametric relationship among the data, which is prevalent in many applications. Further, most notable works in transfer learning and supervised-domain-adaptation focus on classification tasks, and it is unclear how to extend them to regression problems, except for a few works Tolooshams et al. (2021); Lu et al. (2019); Li et al. (2023); Zhang et al. (2024); Chen et al. (2022); Pardoe & Stone (2010). Again, these works treat each domain independently and do not consider any underlying parametric relationship.

In regression tasks such as denoising and deblurring, plug-and-play (PnP) methods Chan (2016); **?** adapt the regularizer by embedding a denoiser into the optimization, making the inference robust to varying data statistics. Our work is related in spirit, as both PnP and TDA introduce adaptivity into model-based inference. The key distinction is that PnP relies on implicit priors through denoisers, whereas TDA introduces explicit, interpretable tunable parameters (such as $\beta$) to handle domain shifts such as noise or gain.

## 3 Domain-Adaptation Through Unrolling

In this section, we discuss the proposed method for adapting a network from the data. In the fine-tuning approach discussed in the previous section, only $\boldsymbol{\beta}$ is learned for the new data. As discussed, in DNNs/CNNs, though there is intuitive reasoning about it, it has not been proven theoretically why only a certain part

of the networks should be tuned. Additionally, the layers of a DNN or a CNN are not interpretable, which makes it challenging to prove the tuning part. Unlike DNNs/CNNs, unrolling (or unfolding)-based methods, which are deep learning models that are derived by unfolding an iterative optimization algorithm into a finite number of layers, are interpretable Gregor & LeCun (2010); Monga et al. (2021). These networks incorporate domain-specific knowledge from optimization techniques and leverage the power of deep learning for efficient and interpretable learning. By noting that the parametric relationship between the domains stems from physical models that generate data and unrolling networks are domain and model-specific, we show that fine-tuning can be made interpretable and efficient.

To elaborate on the interpretable domain adaptation via unrolling framework, let $f_{\boldsymbol{\alpha},\boldsymbol{\beta}}(\cdot)$ be the unfolded network. Unlike standard DNNs/CNNs, the learnable parameters $\{\boldsymbol{\alpha}, \boldsymbol{\beta}\}$ in this network are model-specific and are interpretable. When the network is trained on parametric data $\mathcal{D}_{\boldsymbol{\theta}}$, the network's parameters $\{\boldsymbol{\alpha}, \boldsymbol{\beta}\}$ are function of $\boldsymbol{\theta}$. In the following, we assume that among these, $\boldsymbol{\alpha}$ changes negligibly with $\boldsymbol{\theta}$, whereas, $\boldsymbol{\beta}$ strongly depends on $\boldsymbol{\theta}$. Mathematically, there exist a function $g(\cdot)$ such that

$$\boldsymbol{\beta} = g_{\boldsymbol{\varphi}}(\boldsymbol{\theta}), \tag{4}$$

where $\boldsymbol{\varphi}$ are parameters of the function. The function $g(\cdot)$ can be either derived from the physics of the generating model or learned from the available data. The assumption of the existence of $g_{\boldsymbol{\varphi}}$ is the same as in the fine-tuning approach. However, the assumption holds in practice for many unrolling networks and parametric data, as shown in the following sections. A P-TDA approach is used for known $g$ and $\boldsymbol{\theta}$, to learn the network as

$$\min_{\boldsymbol{\alpha}} \sum_{j=1}^{J} \sum_{(\mathbf{y}_n, \mathbf{x}_n) \in \mathcal{D}_{\boldsymbol{\theta}_j}} d\left(\mathbf{x}_n, f_{\boldsymbol{\alpha}, g_{\boldsymbol{\varphi}}(\boldsymbol{\theta}_j)}(\mathbf{y}_n)\right). \tag{5}$$

The resulting network $f_{\boldsymbol{\alpha}, g_{\boldsymbol{\varphi}}(\boldsymbol{\theta})}(\cdot)$ can then be adapted to test data coming from any domain by substituting appropriate $\boldsymbol{\theta}$. When $g_{\boldsymbol{\varphi}}$ is unknown and $\boldsymbol{\theta}$ is known, it can be learned by solving the problem

$$\min_{\boldsymbol{\alpha}, \boldsymbol{\varphi}} \sum_{j=1}^{J} \sum_{(\mathbf{y}_n, \mathbf{x}_n) \in \mathcal{D}_{\boldsymbol{\theta}_j}} d\left(\mathbf{x}_n, f_{\boldsymbol{\alpha}, g_{\boldsymbol{\varphi}}(\boldsymbol{\theta}_j)}(\mathbf{y}_n)\right). \tag{6}$$

In practice, $g_{\boldsymbol{\varphi}}$ is realized using a neural network where the inputs are the domain information $\boldsymbol{\theta}_j$ as well as the measurements $\mathbf{y}_n$s. In addition, the parameters to the forward model can be used. We observe that the additional information of the measurements improves the performance of the overall tunable network.

In the case of unknown $g$ and $\boldsymbol{\theta}$, we propose the following DD-TDA approach. By noting that the data is implicitly dependent on $\boldsymbol{\theta}$, we train a function $h_{\boldsymbol{\varphi}}(\cdot)$ which is expected to estimate $\boldsymbol{\beta}$ from the data. Specifically, we optimize the following problem

$$\min_{\boldsymbol{\alpha}, \boldsymbol{\varphi}} \sum_{j=1}^{J} \sum_{(\mathbf{y}_n, \mathbf{x}_n) \in \mathcal{D}_{\boldsymbol{\theta}_j}} d\left(\mathbf{x}_n, f_{\boldsymbol{\alpha}, h_{\boldsymbol{\varphi}}(\mathbf{y}_n)}(\mathbf{y}_n)\right). \tag{7}$$

The proposed method results in a single tunable network $f_{\boldsymbol{\alpha}, h_{\boldsymbol{\varphi}}(\cdot)}(\cdot)$. Note that during inference, for any test input $\mathbf{y}$, the network not only predicts the output but also tunes the network via the function $h_{\boldsymbol{\varphi}}$. As in the previous model, the function $h_{\boldsymbol{\varphi}}$ is realized using a neural network.

In the aforementioned formulation, the network $f_{\boldsymbol{\alpha}, h_{\boldsymbol{\varphi}}(\cdot)}(\cdot)$ is trained over the domains $\{\boldsymbol{\theta}_j, j = 1, \cdots, J\}$. The network is expected to adapt well to any data from the trained domains. However, whether it will perform well on an unseen domain $\boldsymbol{\theta} \notin \{\boldsymbol{\theta}_j, j = 1, \cdots, J\}$ depends on the generalization ability of the function $h_{\boldsymbol{\varphi}}(\cdot)$. As we will show in the following sections, the generalization ability depends on the task and how the domain parameter depends on the network parameter. In general, the generalization is good for large $J$, as with any learned network, where the generalization typically improves with the data size.

We would like to emphasize that P-TDA is primarily intended as a *proof-of-concept benchmark* that illustrates the performance upper bound when an ideal domain parameter $\boldsymbol{\theta}$ is available and used to tune $\boldsymbol{\beta}$ via a

mapping. Additionally, we considered comparing P-TDA to cross-validation-based selection of $\boldsymbol{\beta}$, but it is impractical in our setting due to (i) the lack of a well-defined candidate set, and (ii) the need for repeated tuning across domains, which hinders scalability under continuous or high-dimensional shifts. In contrast, DD-TDA learns a data-driven mapping to $\beta$ without requiring access to $\sigma$ or per-domain tuning, making it more suitable for adaptive scenarios with latent or dynamic domain factors. We will clarify these distinctions in the revised manuscript.

In the following sections, we consider three practical scenarios of the aforementioned framework. In particular, we discuss domain adaptation for compressive sensing in the next section, followed by adaptation for blind calibration problems, and finally adaptive phase retrieval.

## 4 Noise-Adaptive Tunable LISTA

This section considers the first problem of domain adaptation through tunable unrolled networks. The problem is to recover sparse vectors from their compressed measurements under different noise conditions. Before detailing the problem and the proposed solution, we first give a quick background of the compressive sensing problem, its solution, and LISTA.

### 4.1 The Compressed Sensing Problem

Consider a set of noisy linear measurements as

$$\mathbf{y} = \mathbf{A}\mathbf{x} + \boldsymbol{\eta}, \quad \boldsymbol{\eta} \sim \mathcal{N}(\mathbf{0}, \sigma^2 \mathbf{I}_{N_y}), \tag{8}$$

where $\boldsymbol{\eta} \in \mathbb{R}^{N_y}$ is an additive Gaussian noise vector with mean $\mathbf{0} \in \mathbb{R}^{N_y}$ and covariance $\sigma^2 \mathbf{I}_{N_y}$. Here, $\mathbf{y} \in \mathbb{R}^{N_y}$ representing the measurement signal, $\mathbf{x} \in \mathbb{R}^{N_x}$ denoting the unknown signal to be estimated, and $\mathbf{A} \in \mathbb{R}^{N_y \times N_x}$ signifying the measurement matrix with $N_y < N_x$. The underdetermined system of equations can be solved with a sparsity assumption on $\mathbf{x}$, that is, $\|\mathbf{x}\|_0 \leq L$ where $L < N_y$. A widely used approach to solve the problem is to minimize the following $\ell_1$-minimization problem:

$$\min_{\mathbf{y}} \|\mathbf{y} - \mathbf{A}\mathbf{x}\|_2^2 + \lambda \|\mathbf{x}\|_1. \tag{9}$$

The objective function encourages solutions that balance data fidelity, captured by the first term, and sparsity, controlled by $\|\mathbf{x}\|_1$. The balance can be achieved by choosing the hyperparameter $\lambda$ appropriately. Several practical algorithms, such as the basis pursuit, iterative hard-thresholding, and ISTA Daubechies et al. (2004); Beck & Teboulle (2009), have been developed to solve the optimization problem.

Most of the aforementioned algorithms are iterative and, starting from an initial estimate, refine the estimate in each iteration. For example, let $\mathbf{x}^k$ be the current estimate of the sparse signal at iteration $k$, $\mathbf{x}^{k+1}$ is the updated estimate for the next iteration. The update rule in ISTA is given as

$$\mathbf{x}^{k+1} = \mathcal{S}_\beta \left( \mathbf{W}_1 \mathbf{y} + \mathbf{W}_2 \mathbf{x}^k \right), \tag{10}$$

with

$$\mathbf{W}_1 = \frac{1}{\kappa} \mathbf{A}^T, \quad \mathbf{W}_2 = \mathbf{I}_{N_x} - \frac{1}{\kappa} \mathbf{A}^T \mathbf{A}, \quad \text{and} \quad \beta = \frac{\lambda}{\kappa}, \tag{11}$$

where $\mathbf{I}_{N_x}$ is the $N_x \times N_x$ identity matrix and $\mathcal{S}_\beta$ is an elementwise soft-thresholding operation with threshold parameter $\beta = \frac{\lambda}{\kappa}$ where $\kappa$ is greater than the largest eigenvalue of $\mathbf{A}^T \mathbf{A}$.

While ISTA and other algorithms work well in practice, their convergence is slow. In addition, the methods are data-independent. On the other hand, given a set of data with some underlying structure (for example, drawn from the same distribution), the algorithms should be able to be fine-tuned to improve the performance. Additionally, LISTA achieves lower recovery error in fewer layers compared to the ISTA with the same number of iterations. This is precisely the LISTA Gregor & LeCun (2010) algorithm where the matrices $\boldsymbol{\alpha} = \{\mathbf{W}_1, \mathbf{W}_2\}$ together with the thresholding parameter $\boldsymbol{\beta}$ are learned from the data by keeping the number of iterations fixed. The LISTA is shown to converge in fewer iterations than the original ISTA.

Before we discuss the proposed noise-adaptive LISTA, we quickly summarize a few results on adaptive LISTA approaches and compare them with the current one.

## 4.2 Variations of LISTA

In the original LISTA framework Gregor & LeCun (2010), the network is trained to learn the parameters $\{\mathbf{W}_1, \mathbf{W}_2, \boldsymbol{\beta}\}$ (cf. equation 11) using data generated from a specific distribution and a known measurement matrix $\mathbf{A}$. Changes in $\mathbf{A}$ lead to changes in the distribution, affecting the measurements $\mathbf{y}_n$ and necessitating network retraining. To address this, Aberdam et al. (2021) introduced an adaptive LISTA algorithm where matrices $\mathbf{W}_{1,2}$ are decomposed into two parts: a fixed component dependent on $\mathbf{A}$ and a learnable component similar to traditional LISTA. This enables training on data generated by multiple choices of $\mathbf{A}$, optimizing the network accordingly. At inference, the network adapts to new $\mathbf{A}$ before prediction, demonstrating improved performance across varying $\mathbf{A}$ values with theoretical guarantees. However, this method requires precise domain parameter knowledge (here $\mathbf{A}$) during inference, and the matrix decompositions introducing domain dependence are less straightforward, leaving room for alternative approaches Chen et al. (2018); Liu & Chen (2019); Chen et al. (2021). While our focus differs, our DD-TDA framework does not necessitate prior domain parameter knowledge. In cases where it is known, we optimize its use akin to the P-TDA approach.

Another line of work reduces reliance on learning. Analytic-LISTA Liu & Chen (2019) shows that $\mathbf{W}_{1,2}$ can be analytically derived from $\mathbf{A}$, eliminating the need to learn them. Li *et al.* Li et al. (2024) proposed an error-based thresholding scheme, where the layer-wise threshold is set as a function of the reconstruction error. Chen *et al.* Chen et al. (2021) further enhanced LISTA by adding momentum, yielding practical rules for adaptively calculating layer parameters and improving both generalization and training efficiency.

While these approaches introduce adaptivity at the algorithmic or parameter level, our focus is complementary: the proposed DD-TDA framework learns to adjust domain-sensitive parameters *without requiring prior knowledge of the domain*, while P-TDA leverages domain knowledge when it is available.

## 4.3 Noise-Level-Specific Domains

In this problem, noise variance parameterized the domains, and the domain-specific measurements are given as

$$\mathcal{D}_{\theta_j = \sigma_j} = \{\mathbf{y}_{ij}, \mathbf{x}_i\}_{i=1}^N \ \forall \ j = \{1, 2, \ldots, J\}, \tag{12}$$

where

$$\mathbf{y}_{ij} = \mathbf{A}\mathbf{x}_i + \boldsymbol{\eta}_j, \quad \boldsymbol{\eta}_j \sim \mathcal{N}(\mathbf{0}, \sigma_j^2 \mathbf{I}_{N_y}), \tag{13}$$

and $\mathbf{I}_{N_y}$ is an identity matrix of dimension $N_y \times N_y$

Specifically, for each domain, the noise levels are different while the rest of the measurement model remains fixed. The measurement model is practically relevant, as the noise levels can vary significantly. As discussed in Section 2, both the PT and JT approaches have limitations. Here, we consider the problem of fine-tuning the LISTA model to adapt it to different noise domains.

For ease of discussion, the different domains, based on the noise levels, are also characterized by average SNR levels defined as

$$\mathrm{SNR}_j = \frac{1}{N} \sum_{i=1}^N 10 \log_{10} \left( \frac{\frac{1}{N_y} \|\mathbf{A}\mathbf{x}_i\|_2^2}{\sigma_j^2} \right). \tag{14}$$

## 4.4 A Retraining Approach

Before introducing our noise-tunable LISTA framework, we first examine the effectiveness of a retraining-based method that utilizes a DNN to address the noise-tunability challenge. The experiment is conducted on a dataset with three domains with SNRs of 6 dB, 16 dB, and 32 dB. Data is generated following equation 12 and equation 13 (see the data generation part of Section 4.5 for details). We start by training a fully connected neural network on a combined dataset containing samples from all domains. The network consists of three hidden layers with 256, 512, and 256 neurons, respectively, each followed by ReLU activations. For

Table 1: Test results of retrained DNNs for domain adaptation

| Training approach | D1(6 dB) | D2(16 dB) | D3(31 dB) |
|---|---|---|---|
| | NMSE (in dB), HR(%) | | |
| Joint train | -6.60, 54.22 | -12.34, 76.72 | -13.33, 79.81 |
| Retrain last layer | -6.72, 53.73 | -12.64, 77.87 | -13.79, 81.17 |
| Retrain last 3 layers | -7.10, 53.93 | -13.92, 83.40 | -16.17, 89.24 |

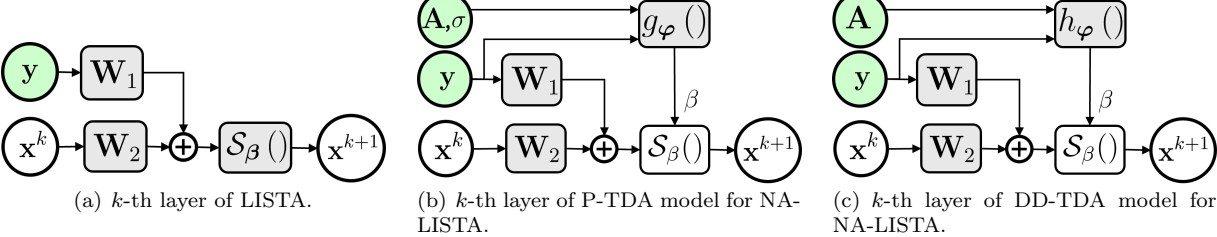

(a) $k$-th layer of LISTA.  (b) $k$-th layer of P-TDA model for NA-LISTA.  (c) $k$-th layer of DD-TDA model for NA-LISTA.

Figure 1: Layers/iterations of different LISTA architectures: (a) Conventional LISTA, (b), (c) NA-LISTA. (b) Single layer of the P-DTDA model with input data $\{\mathbf{y}, \mathbf{A}\}$ and domain parameter $\boldsymbol{\sigma}$ available during inference. (c) Single layer of DD-TDA model with only input data $\{\mathbf{y}, \mathbf{A}\}$ available during inference. Note: Green blocks represent the data available at inference time, and Gray blocks are trainable parameters.

domain-specific adaptation, we perform staged retraining by progressively unfreezing deeper layers of the network on the target domain data. For retraining, we use domain-specific data that was previously included in a mixed-domain dataset during the initial training phase. Initially, only the final layer is retrained; then, we freeze the first hidden layer and fine-tune the remaining layers, including the output layer. Results in Table 1 show that retraining just the final layer does not yield significant improvements across varying noise levels. However, as more layers are made trainable, performance on the test data, measured in terms of NMSE and HR (defined in equation 16 and equation 17), improves consistently. These findings suggest that adapting only the final layer is insufficient for effective tunability. Instead, identifying the optimal subset of layers to retrain may require a trial-and-error approach, which can be task-specific and assumes access to domain-specific data for adaptation.

### 4.5 Proposed Noise-Adaptive LISTA (NA-LISTA)

Unlike the previous DNN-based methods, we show that LISTA is tunable to noise levels. Specifically, we note that among the learnable parameters of LISTA, $\boldsymbol{\alpha} = \{\mathbf{W}_1, \mathbf{W}_2\}$ and $\beta$, the latter one is a strong function of the noise variance $\sigma$ for a fixed $\mathbf{A}$, that is, we have that $\beta = g(\sigma)$. There are several choices of $g$ Galatsanos & Katsaggelos (1992) and one of the widely used ones is given as Donoho & Johnstone (1994)

$$\beta = \sigma\sqrt{2\log N_x}/\kappa. \tag{15}$$

The above formulation implies that the higher the noise variance, the larger the soft thresholding is. Since the soft-thresholding operation promotes sparsity, a larger $\beta$ generates sparser estimates. The choice in equation 15 requires precise knowledge of $\sigma$, which may not be available in practice and has to be estimated using different methods Galatsanos & Katsaggelos (1992); Liu et al. (2008; 2012). An alternative approach is cross-validation Golub et al. (1979) where $\beta$ is learned from validation data. The learned parameter is not optimal if the validation data consists of measurements with various noise levels.

To address the limitations mentioned above, we apply the proposed P-TDA (assuming a known $\sigma$) and DD-TDA methods. One iteration or layer of these two methods, including the conventional LISTA, are shown in Fig. 1. In DD-TDA, $\{\mathbf{A}, \mathbf{y}, \sigma\}$ are used to tune the network by altering $\beta$, whereas, in DD-TDA framework, only $\{\mathbf{A}, \mathbf{y}\}$ are given as input to $h_{\boldsymbol{\varphi}}$ to tune $\beta$. In the following, we discuss the data generation and results.

**Data Generation and Network Architecture**: To evaluate the proposed NA-LISTA, we generated a synthetic dataset as per equation 12–equation 13. The matrix $\mathbf{A}$ was kept fixed in all the experiments, and its entries were sampled from a standard normal distribution followed by columnwise normalization. Further, we set $N_x = 100$, $N_y = 30$, and $L = 3$. The entries of the noise vector were zero-mean Gaussian random variables with domain-specific standard deviations. Specifically, noise variance $\sigma_j$ simulates $J$ distinct SNR levels for robustness assessment. The total number of samples generated was split into train, validation, and test sets in the ratio of $56\%, 24\%$, and $20\%$, respectively.

Since samples from all the domains were used to train and test the JT, P-TDA, and DD-TDA methods in order to maintain dataset size uniformity, we used the same number of samples for each PT model. For example, for the $J$ domains used for training, we generated $N_{train}$(training set slice of $N$) examples from each domain, and used $N_{train} \times J$ training samples for JT, P-TDA, and DD-TDA methods. Whereas, we generated $N_{train} \times J$ examples for training each domain-specific PT model. In the following experiments, we consider $J = 3$ and $N = 43,000$.

LISTA networks with the proposed $g_{\boldsymbol{\varphi}}$ and $h_{\boldsymbol{\varphi}}$ networks were jointly trained in an end-to-end fashion using backpropagation, and the mean-squared error (MSE) loss was used as the cost function. The MSE was measured by averaging the Euclidean distance $\|\mathbf{x}^* - \hat{\mathbf{x}}\|_2^2$ over the training batch, where $\mathbf{x}^*$ and $\hat{\mathbf{x}}$ denote the ground truth and predicted sparse signals, respectively. The network parameters were initialized as mentioned in equation 11, where $\lambda$ was initialized to 0.1 and $\kappa$ was 1.001 times the maximum absolute eigenvalue of $\mathbf{A}^T\mathbf{A}$. Further, in our experiments, all methods—PT, JT, P-TDA, and DD-TDA—assume that the sensing matrix $\mathbf{A}$ is known and fixed, ensuring a fair comparison. In all the methods, $\mathbf{W}_1$ and $\mathbf{W}_2$ are initialized using $\mathbf{A}$.

**Performance Metrics**: Model performance was evaluated using Normalized Mean Squared Error (NMSE) and Hit Rate (HR). Let $\mathbf{x}^*$ be the ground truth signal and $\hat{\mathbf{x}}$ be the predicted sparse signal. The NMSE contribution by each test sample is given by

$$\text{NMSE} = \frac{\|\mathbf{x}^* - \hat{\mathbf{x}}\|_2^2}{\|\mathbf{x}^*\|_2^2}. \tag{16}$$

For our experiments, the average NMSE over the entire test set was computed and reported in decibels.

The HR, with a relative error tolerance $t$, is defined as

$$\text{HR} = \frac{\sum_{n=1}^{N_x} \mathbb{1}(\mathbf{x}^*(n) \neq 0)\mathbb{1}(|\mathbf{x}^*(n) - \hat{\mathbf{x}}(n)| \leq t\,\mathbf{x}^*(n))}{L}. \tag{17}$$

Here, $n$ indexes the elements of the vectors $\mathbf{x}^*$ and $\hat{\mathbf{x}}$. HR measures the fraction of nonzero entries in $\hat{\mathbf{x}}$ whose estimates fall within a relative error tolerance of $t$, set to 0.3 in our experiments. For our experiments, the average HR (in %) over the entire test set is reported.

**Experiments**: We conducted three evaluations. The first one focused on a broad range of SNRs, another covers a lower SNR range, and a third for testing generalization by evaluating on unseen SNR levels. The first two experiments were designed to test whether the proposed method can generalize across domains with large variations in noise level, as well as when domain shifts are subtle and more challenging to distinguish. In each of these, our proposed DD-TDA and P-TDA methods were compared with the baseline JT and PT methods. For both JT and PT, conventional LISTA networks were trained.

**Comparison with the existing methods:** For the broad range we considered three noise levels as D1 ($\sigma = 0.1$, SNR = 6 dB), D2 ($\sigma = 0.03$, SNR = 16 dB), and D3($\sigma = 0.005$, SNR = 32 dB). For the narrow range, the SNRs are given as D1 ($\sigma = 0.12$, SNR = 4 dB), D2 ($\sigma = 0.06$, SNR = 10 dB), and D3 ($\sigma = 0.035$, SNR = 15 dB). For these two scenarios, the NMSEs and HRs of the methods when tested for each domain are shown in Figs. 3(d) and 3(f). Here, PT-D1, PT-D2, and PT-D3, refers to PT methods trained for domains D1, D2, and D3, respectively. We observe that PT methods trained for a specific domain generally perform well when tested on the same domain. Specifically, the observation is true in terms of NMSE, but not always in terms of HR, as it is a stricter metric and can exhibit more variability. The performance deteriorates when trained and tested on different domains. Further, PT gives lower error (better HR) than

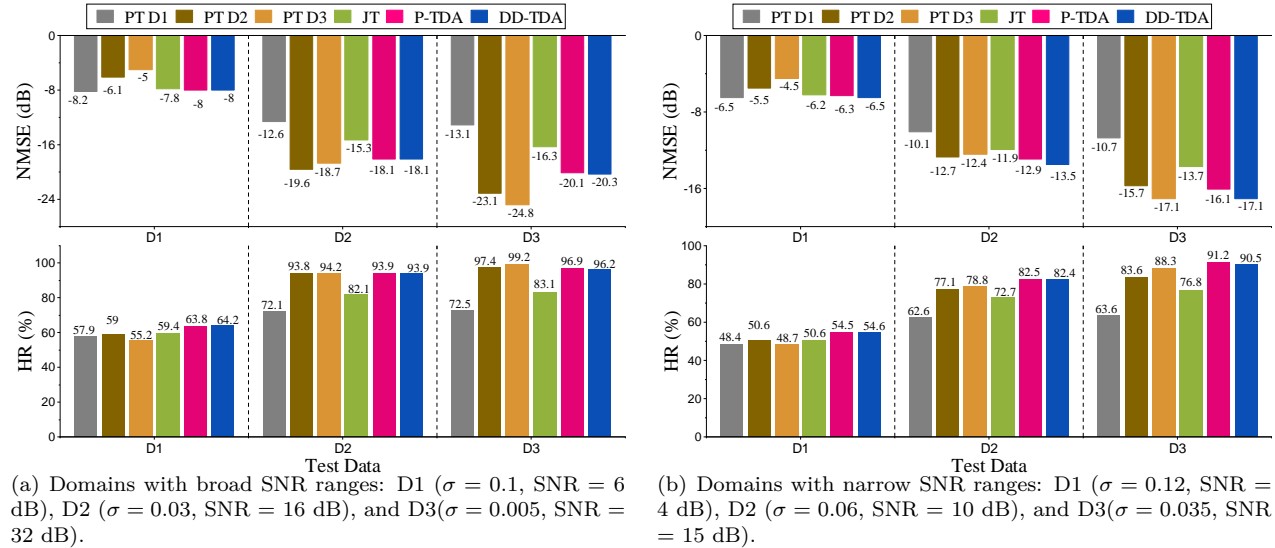

(a) Domains with broad SNR ranges: D1 ($\sigma = 0.1$, SNR = 6 dB), D2 ($\sigma = 0.03$, SNR = 16 dB), and D3($\sigma = 0.005$, SNR = 32 dB).

(b) Domains with narrow SNR ranges: D1 ($\sigma = 0.12$, SNR = 4 dB), D2 ($\sigma = 0.06$, SNR = 10 dB), and D3($\sigma = 0.035$, SNR = 15 dB).

Figure 2: Comparison of the proposed methods with the JT and PT methods for NA-LISTA for different SNR ranges. PT's performances are better than JT's for each domain. The P-TDA/DD-TDA has comparable performance to PT, demonstrating the noise-tunability aspect.

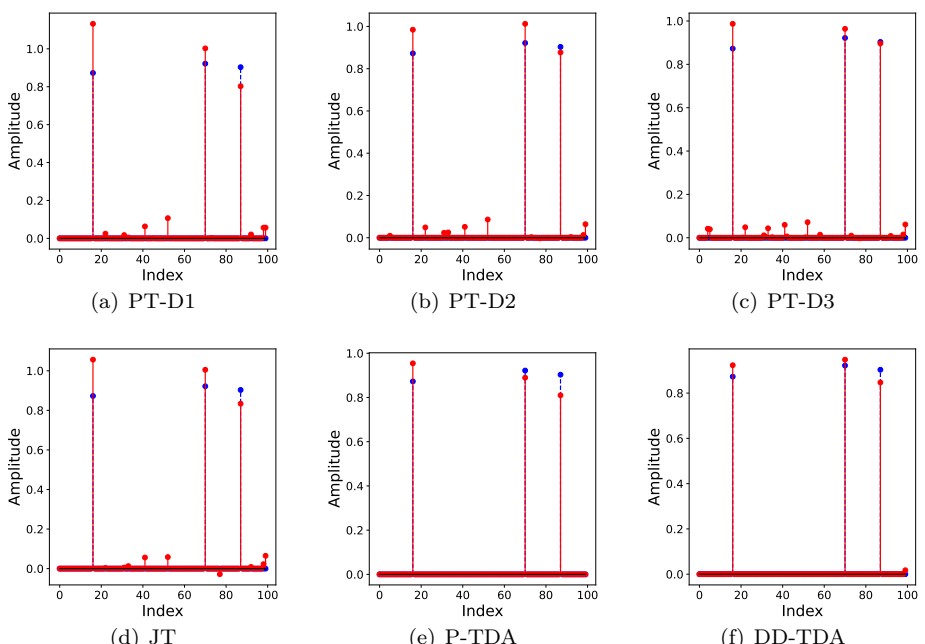

Figure 3: Visual comparison of reconstructed signals on D2 test data for the broad SNRs scenario. The figure illustrates how model variations impact reconstruction quality, highlighting differences in signal fidelity and robustness across approaches.

JT for all the domains, and this observation is more evident in the HRs, where PT shows a performance improvement of $6 - 12\%$ over the JT for D2 and D3. We note that the JT and PT perform similarly on D1, likely because PT-D1 was trained only on highly noisy samples, unlike JT, which saw both clean and noisy data, leading to PT-D1 underperforming relative to expectations.

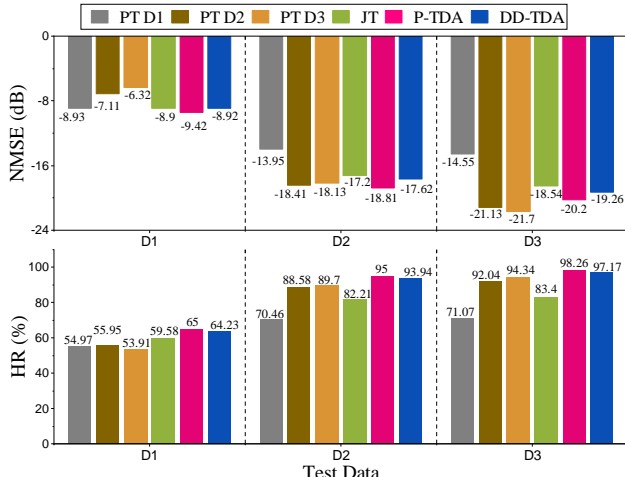

Figure 4: Comparison of the proposed methods with the JT and PT methods for NA-LISTA for broad SNR ranges, averaged over 5 experiments.

Comparing the proposed P-TDA and DD-TDA with the PT method, we observe that the proposed approaches are performing on par with the respective PT methods for each domain. Focusing on the DD-TDA method, which does not consider the domain knowledge, and has comparable performance with the PT methods. The only notable performance gain of PT over our methods is that in D3 during the broad SNR range experiment, with an NMSE gain of around 5 dB, likely due to PT-D3 having access to significantly more clean training examples than the DD-TDA method. In Fig. 3, we now provide an example reconstruction from the D2-domain for visual comparison. The plots clearly illustrate the estimation accuracy and highlight the improvements achieved by the proposed methods, complementing the quantitative results.

Further, DD-TDA occasionally matches or even outperforms P-TDA due to: (i) its end-to-end task-specific learning of $\beta$, (ii) the limited flexibility of P-TDA's fixed mapping from $\sigma$ to $\beta$, (iii) implicit regularization effects in DD-TDA that enhance robustness, and (iv) minor stochastic variations during training. To understand these variations better, in Fig. 13, we show the metrics averaged over five repetitions of the experiments from a broad-SNR range. We note that, with the repeated experiments, the observations are consistent with the previous experiments.

In a nutshell, the results show the tunability aspect of the NA-LISTA for different noise levels.

Having assessed the fact that few network parameters are tunable with the domains, next, we show that the remaining parameters are nearly invariant to the domains.

**Invariance of $\boldsymbol{\alpha}$ with $\theta$:** Here, we asses the variation of the network parameters $\boldsymbol{\alpha} = \{\mathbf{W}_1, \mathbf{W}_2\}$ and threshold $\beta$ with the noise levels $\theta = \sigma$. For NA-LISTA, it was assumed that $\boldsymbol{\alpha}$ does not vary significantly with $\sigma$, and $\beta$ changes significantly with domains. To further demonstrate invariance of $\mathbf{W}_{1,2}$ with domain shifts, in Fig. 5 showed $\mathbf{W}_1$ for different domains, and Figs. 6 and 7, shows zoomed-in versions of the matrices of better visualization. The images show that the values of the matrices do not vary significantly with the domains.

To assess the invariance of $\boldsymbol{\alpha}$ across domains, we define the normalized Frobenius norm metric. To compare the structure of learned weight matrices across trained models, we define a scale-invariant metric such that it normalizes the Frobenius norm by the maximum absolute value in the matrix. Mathematically, we measured the metric $S(\mathbf{W}) = \frac{\|\mathbf{W}\|_F}{\max |\mathbf{W}|}$ and the values are shown in Table 2. The division by the maximum entry was used only as a simple normalization (in the same spirit as NMSE), so that the values are on a comparable scale. We note negligible changes in the metric for both matrices when the networks are trained for different domains. On the other hand, the values of $\beta$ have a significant change with domains, justifying the assumption.

Table 2: $S(\mathbf{W})$ and $\beta$ values for PT models

| Domains (SNR) | D1 (6 dB) | D2 (16 dB) | D3 (32 dB) |
|---|---|---|---|
| $S(\mathbf{W}_1)$ | 16.35 | 16.1 | 15.55 |
| $S(\mathbf{W}_2)$ | 15.07 | 17.77 | 15.62 |
| $\beta$ | 0.088 | 0.048 | 0.037 |

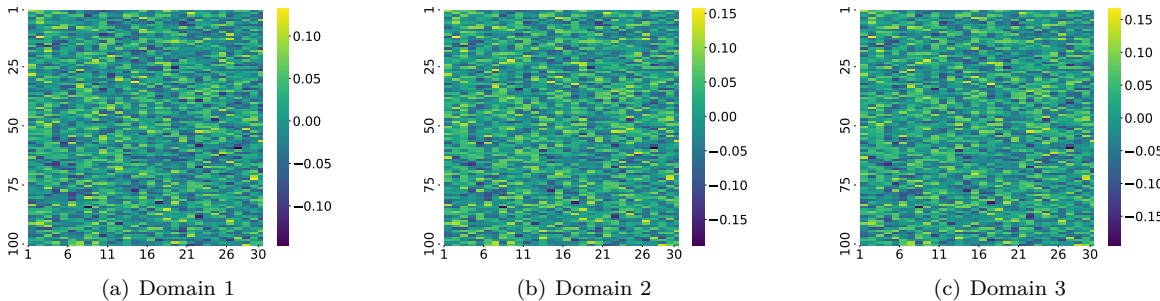

    (a) Domain 1             (b) Domain 2             (c) Domain 3

Figure 5: Comparison of learned $\mathbf{W}_1$ matrices for different domains.

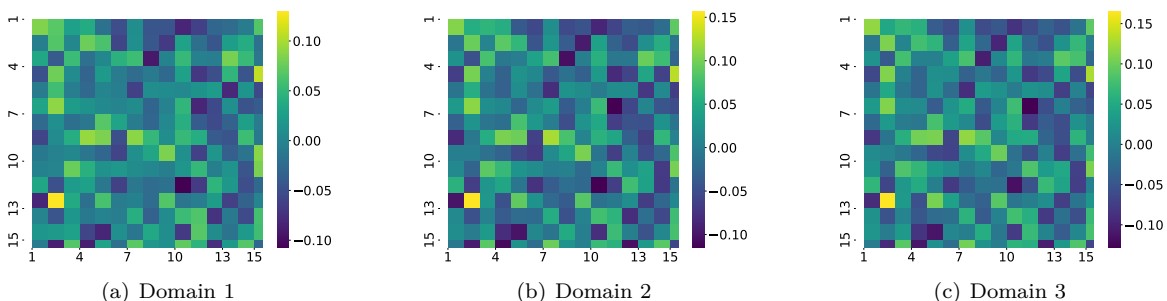

    (a) Domain 1             (b) Domain 2             (c) Domain 3

Figure 6: Comparison of top $(15 \times 15)$ of learned $\mathbf{W}_1$ matrices.

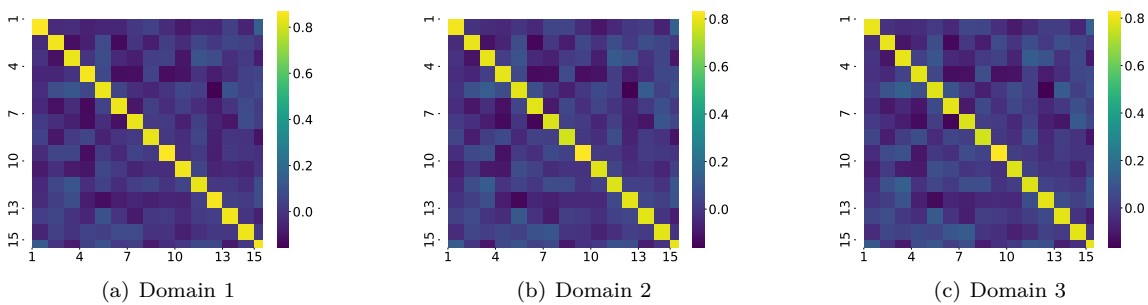

    (a) Domain 1             (b) Domain 2             (c) Domain 3

Figure 7: Comparison of top $(15 \times 15)$ of learned $\mathbf{W}_2$ matrices for different domains.

Next, we discuss the generalization ability of the proposed approaches.

**Generalization:** Building on the previous experiments, which evaluated the model's performance on domains with known SNR levels, we now assess its ability to generalize to unseen domains. Unlike before, the

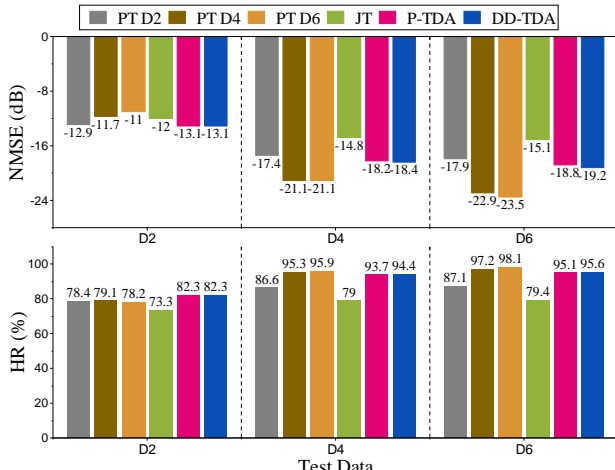

Figure 8: Results for generalization experiment: The models (except PT) were trained on D1, D2, and D3 domains. Observation: DD-TDA generalizes well and performs better than JT models.

model is tested on SNR levels that were not part of the training set, allowing us to evaluate its adaptability and robustness in entirely new noise conditions. We considered six domains, D1-D6, with average SNRs (in dB) of $\{4, 10, 15, 20, 26, 32\}$. The domain-specific PT models were trained on D2, D4, and D6 datasets with $N$ samples each. All the mix domain models, JT, DD-TDA, and P-TDA, were trained on a mix dataset of the same $N$ samples from domains D1, D3, and D5, while they were tested on D2, D4, and D6.

Since this experiment evaluates the ability to generalize to unseen domains, we compare our proposed models specifically against the JT method, focusing on their respective generalization capabilities. The results, presented in Fig. 8, demonstrate that our methods outperform the baseline JT approach, especially in the HR metric by around 9-16%. On comparing the P-TDA and DD-TDA methods with PT methods, we observe that PT methods result in a lower error of $3 - 5$ dB for D4 and D6, which are high SNR domains. Whereas, for D2, with SNR of 10 dB, the P-TDA and DD-TDA have comparable performance to the PT approach.

In the following, we consider another application for the P-TDA and DD-TDA methods.

## 5 Domain-Adaptive Sparse Gain Calibration

Consider the measurements of the form

$$\mathbf{y} = \mathrm{diag}(\mathbf{c})\mathbf{A}\mathbf{x} + \boldsymbol{\eta}, \tag{18}$$

where, except $\mathbf{c} \in \mathbb{R}^{N_y}$, the rest of the parameters have the same dimensions and characteristics as in equation 8. Here $\mathbf{c}_i$ is the gain of $\mathbf{y}_i$-th measurement. In particularly, $\mathbf{x}$ is $L$-sparse. The measurement model is prevalent in several applications, such as time-of-flight imaging Steiger et al. (2008); Mersmann et al. (2013), direction of arrival estimation with unequal sensor gains Paulraj & Kailath (1985), and more. An unknown $\mathbf{c}$ amounts to a sparse-blind calibration problem Schulke et al. (2013). In Tolooshams et al. (2023), a data-driven approach, based on LISTA, was proposed to solve for an unknown but fixed $\mathbf{c}$. However, the network has to vary as $\mathbf{c}$ varies, which is the practice case. Here, we consider the applicability of the proposed tunability approach by considering the domain parameter as $\mathbf{c}$. Specifically, the data for the parametric domains are defined as

$$\mathcal{D}_{\boldsymbol{\theta} = \mathbf{c}_j} = \{\mathbf{y}_{ij}, \mathbf{x}_i\}_{i=1}^N \; \forall \; j = \{1, 2, \dots, J\}, \tag{19}$$

where

$$\mathbf{y}_{ij} = \mathrm{diag}(\mathbf{c}_j)\mathbf{A}\mathbf{x}_i + \boldsymbol{\eta}_j, \quad \boldsymbol{\eta} \sim \mathcal{N}(\mathbf{0}, \sigma^2 \mathbf{I}_{N_y}). \tag{20}$$

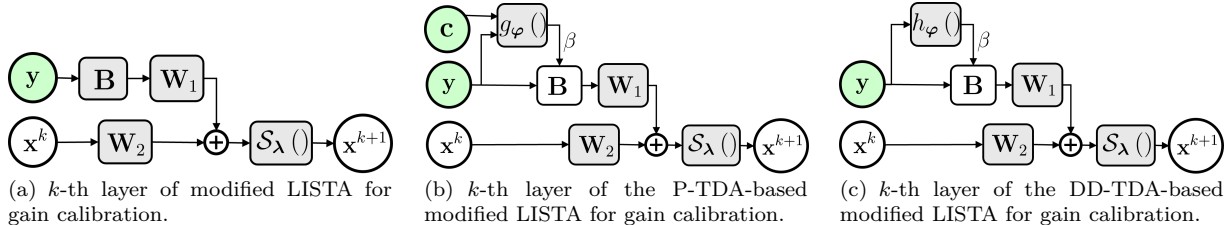

(a) $k$-th layer of modified LISTA for gain calibration.

(b) $k$-th layer of the P-TDA-based modified LISTA for gain calibration.

(c) $k$-th layer of the DD-TDA-based modified LISTA for gain calibration.

Figure 9: Tunable model for blind calibration gain with **B** as a tuned parameter.

Note that the noise variance is constant across domains, unlike the parametric-domain model in equation 12 and equation 13. However, this is not a limitation; noise ($\boldsymbol{\eta}$) and gain (**c**) could be the domain parameters. Since our objective is to show the tunability aspect of the network as **c** varies, $\sigma$ is kept fixed. Next, we discuss the details of the data generation and network architecture, followed by the simulation results.

### 5.1 The Proposed Approaches and Network Architecture

We first propose an approach that is applicable to both the PT and JT methods. Then, we suggest a modification to make the method domain-adaptive.

There are several possible approaches to solving the sparse-gain calibration problem. For example, one could treat $\text{diag}(\mathbf{c})\mathbf{A}$ as the measurement matrix and then apply the conventional LISTA algorithm. Alternatively, one could learn the matrices $\text{diag}(\mathbf{c})$ and $\mathbf{A}$ separately, as in Tolooshams et al. (2023). The former approach lacks adaptability, as the matrix $\text{diag}(\mathbf{c})$ is not learned independently. In contrast, the latter approach leads to a more complex network, since the information in $\text{diag}(\mathbf{c})$ is embedded within each layer of LISTA.

Another alternative, when **c** is known, is to solve the optimization problem in equation 8 using LISTA, after normalizing the measurements via $\hat{\mathbf{y}} = \text{diag}(\mathbf{c}^{-1})\mathbf{y}$. However, this may significantly amplify noise in measurements corresponding to small values of **c** — a well-known issue in deconvolution problems. Moreover, the exact values of **c** may not be known in practice.

We address these issues by proposing an approach based on Wiener filtering for deconvolution, combined with a learning strategy. Specifically, instead of using the normalized measurements $\text{diag}(\mathbf{c}^{-1})\mathbf{y}$, we learn a vector **b** jointly with the LISTA parameters, such that $\text{diag}(\mathbf{b})\mathbf{y}$ is used as the input to LISTA. As in Wiener filtering, the entries of the learned **b** are expected to be small when the SNR for the corresponding measurement is low. This approach mitigates the problems associated with directly using $\mathbf{c}^{-1}$. Furthermore, deconvolution is performed only once, after which the subsequent layers follow the conventional LISTA structure (cf. Fig. 9(a)). The network can be trained either jointly across all domains or individually per domain to learn the parameters $\mathbf{b}, \mathbf{W}_{1,2}$ and the soft-thresholding parameter $\boldsymbol{\lambda}$.

Since the domains are parameterized by the calibration vector **c**, enabling adaptability or tunability requires identifying the parameters that vary significantly with **c**. In Fig. 9(a), the matrix **B** is introduced to compensate for the gain factor, while the rest of the network retains the structure of the traditional LISTA. Moreover, in the noise-free case, if one were to solve the problem using conventional ISTA, **B** would act as $\mathbf{c}^{-1}$. Hence, the calibration gains $\boldsymbol{\beta} = \mathbf{B}$ are expected to vary significantly with **c**, in contrast to the parameters $\boldsymbol{\alpha} = \mathbf{W}_{1,2}, \lambda$, which are relatively invariant.

Motivated by this insight, we propose the P-TDA and DD-TDA methods, as shown in Figs. 9(b) and (c), respectively. In the P-TDA method, **B** is learned from both the known **c** and the measurements using the network $g_{\boldsymbol{\varphi}}()$. In contrast, in the DD-TDA approach, **B** is inferred solely from the measurements using the network $h_{\boldsymbol{\varphi}}()$.

**Data Generation**: For the experiment, we considered two sets of parametric domain datasets following the model in equation 19. The first set uses structured gains, where **c** follows a sinusoidal distribution.

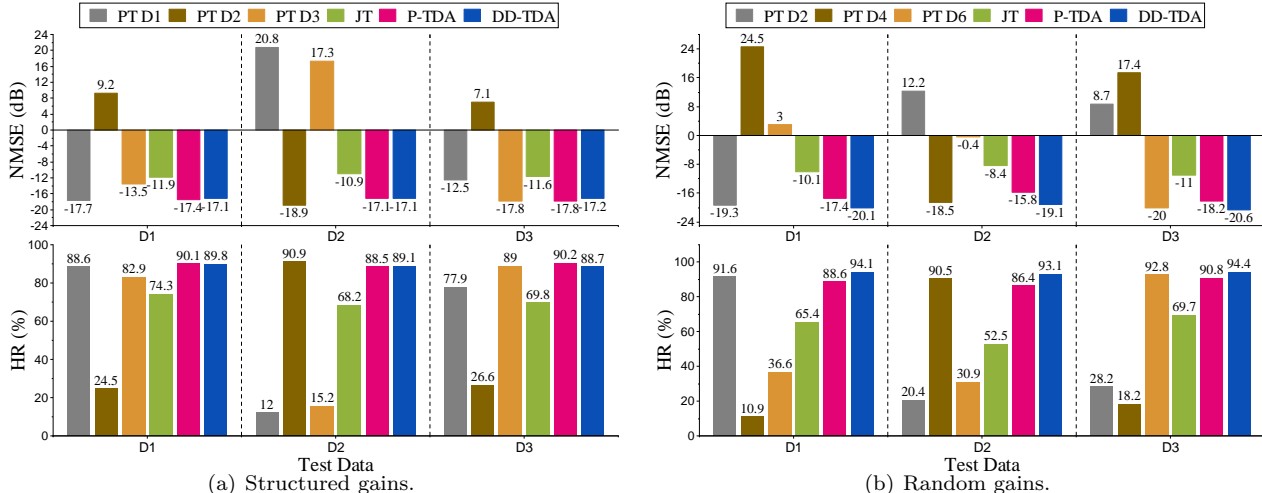

Figure 10: Comparison of the proposed methods with the JT and PT methods for the sparse gain calibration problem. The P-TDA/DD-TDA has comparable performance to PT, demonstrating the domain-adaptability aspect.

Specifically, the entries of $\mathbf{c}$ for each domain were generated as

$$\mathbf{c}_j(n) = |a \sin(2\pi f_j n + \phi_j) + b|. \tag{21}$$

The variation in $\mathbf{c}_j$ across domains is governed by changes in $\{f_j, \phi_j\}$ and can therefore be controlled. In contrast, when $\mathbf{c}_j(n)$ was randomly generated, the resulting measurements vary significantly across domains.

In the experiments, we fixed $a = 0.5$ and $b = 0.6$, and sampled $f_j, \phi_j$ uniformly at random from the interval $(0, 1]$. For the random gain scenario, each $\mathbf{c}_j(n)$ was sampled independently from a uniform distribution over $[0.1, 1.3]$. In both the structured and random gain cases, we considered three domains ($J = 3$) and generated $N = 43,000$ examples per domain, following the procedure in Section 4.5.

The training procedures for JT, PT, P-TDA, and DD-TDA were identical to those described in Section 4.

Using the setup, we evaluated the methods in terms of NMSE and HR. The results are shown in Figs. 10(a) and 10(b). The following observations can be made. For both gain models, the JT approach results in NMSEs that are 5–10 dB higher than those achieved by the corresponding PT approach. In the PT method, training and testing on different domains yields substantially higher errors (more than 20 dB NMSE) compared to NA-LISTA (cf. Figs. 3(d) and 3(f)). These cross-domain errors are particularly severe when measurement changes are significant across domains due to variations in the domain parameter.

To quantify domain similarity, we compute the cosine similarities among the vectors $\{\mathbf{c}_j\}_{j=1}^3$. Let $\rho_{i,j}$ denote the similarity between $\mathbf{c}_i$ and $\mathbf{c}_j$. For the structured and random gain experiments, the similarity values are $\{\rho_{1,2}, \rho_{2,3}, \rho_{3,1}\} = \{0.73, 0.76, 0.99\}$ and $\{0.59, 0.78, 0.82\}$, respectively. Due to the high similarity between domains 1 and 3 in the structured gain case ($\rho_{1,3} = 0.99$), the NMSE degradation when testing PT-D1 on D3 and vice versa is limited to around 5 dB. In contrast, for less similar pairs —for example, testing PT-D2 on D3— the NMSE degradation exceeds 20 dB.

Comparing the proposed P-TDA and DD-TDA methods to PT, we find that the NMSE and HR values remain consistent across domains. These results demonstrate that a single network can adapt to multiple domains, eliminating the need to train separate networks for each one.

Next, we examine the generalization capability of our proposed methods.

**Generalization:** To assess the domain generalization capability of the proposed approaches for the calibration problem, we trained the models on 45 different domains (D1-D45) and then tested them on five unseen

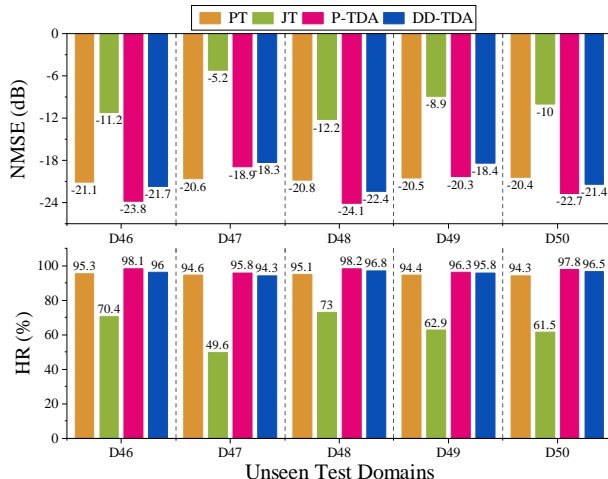

Figure 11: Generalization performance of models, across domains with structured calibration gains. The JT, P-TDA, and DD-TDA models were trained on 45 domains and tested on 5 unseen domains.

domains (D46-D50). For the JT, P-TDA, and DD-TDA methods, we generated $N = 10,000$ examples from each domain, which amounts to a total of $500,000$ examples for training and testing. Whereas, for the PT models, we generated 50000 examples per domain. The datasets were divided into test, validation, and train sets in proportions of $10\%, 20\%,$ and $70\%$, respectively. In all these cases, the gains were generated using equation 21.

The NMSEs and HRs, for the JT and PT methods, together with P-TDA and DD-TDA, are shown in Fig. 11. We note that for all the unseen domains, the NMSEs in the PT approaches are 8-15 dB lower than the JT methods. Both the P-TDA and DD-TDA methods result in 10-13 dB lower NMSEs compared to the JT methods. Comparing the P-TDA and the PT methods, we note that the NMSEs of the P-TDA approach are within a $\pm 3$ dB range of the corresponding PT error for all the unseen domains. Similarly, the HRs of P-TDAs are within $\pm 4\%$ of those of the PT. This shows the generalization ability of the proposed networks.

# 6 Domain Adaptive Phase Retrieval Algorithm

We next demonstrate the proposed tunable domain adaptation framework on the sparse phase retrieval (PR) problem Wang et al. (2017); Naimipour et al. (2024); Jian-Feng et al.. Specifically, the measurements are modeled as

$$\mathbf{y} = |\mathbf{A}\mathbf{x}|^2 + \boldsymbol{\eta}, \quad \boldsymbol{\eta} \sim \mathcal{N}(\mathbf{0}, \sigma^2 \mathbf{I}_{N_y}), \tag{22}$$

where $\mathbf{y} \in \mathbb{R}^{N_y}$, $\mathbf{x} \in \mathbb{R}^{N_x}$, $\mathbf{A} \in \mathbb{R}^{N_y \times N_x}$ with $\|\mathbf{x}\|_0 = L$, and $\boldsymbol{\eta} \in \mathbb{R}^{N_y}$ is Gaussian noise. Unlike the linear settings considered earlier, here only the magnitudes of the measurements are observed. The goal is to estimate $\mathbf{x}$ (up to a global phase), which makes the measurements a nonlinear function of $\mathbf{x}$. We show that a variant of LISTA can not be applied in this case.

Conventional sparse PR algorithms typically assume that the sparsity level $L$ is known. If $L$ changes, both traditional algorithms Wang et al. (2017); Jian-Feng et al. and their unrolled counterparts Naimipour et al. (2024) may fail. Hence, we consider the sparsity $L$ as the domain parameter, that is, $\theta = L$. In what follows, we first review an unrolled solution to the PR problem used for PT and JT, then identify the tunable parameter $\beta$, and finally describe the design of $g_{\boldsymbol{\varphi}}$ and $h_{\boldsymbol{\varphi}}$ for P-TDA and DD-TDA.

## 6.1 Phase Retrieval Algorithm

We build on the sparse-PR algorithm proposed in Jian-Feng et al., which consists of two stages. The first stage estimates the support of $\mathbf{x}$ by minimizing $C_A(\mathbf{x}) = \frac{1}{2N_y}\|\mathbf{z} - |\mathbf{A}\mathbf{x}|\|_2^2$, where $\mathbf{z}$ is the elementwise

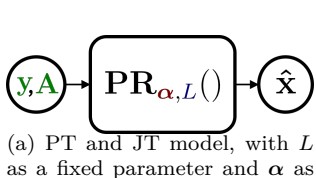
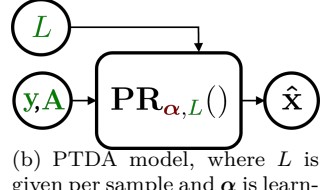
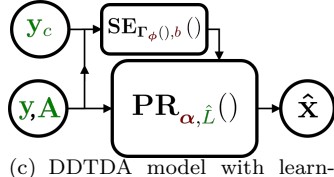

(a) PT and JT model, with $L$ as a fixed parameter and $\boldsymbol{\alpha}$ as learnable parameter.

(b) PTDA model, where $L$ is given per sample and $\boldsymbol{\alpha}$ is learnable parameter.

(c) DDTDA model with learnable paramters $\{\boldsymbol{\phi}, b, \boldsymbol{\alpha}\}$, where $\hat{L}$ is per sample estimated sparsity.

Figure 12: $\mathbf{PR}_{\boldsymbol{\alpha},L}()$ is the tunable model for sparse phase retrieval. Note: Green symbols represent the data available at inference time. Red symbols are trainable parameters, and blue symbols are fixed parameters.

square-root of $\mathbf{y}$. At the $(k+1)$-th iteration, the support is estimated as

$$S_{k+1} = \mathrm{supp}\big(\mathcal{H}_L\big(\mathbf{x}^k - \alpha_{1,k}\nabla C_A(\mathbf{x}^k)\big)\big), \tag{23}$$

where $S_{k+1} \subset \{1, 2, \cdots, N_x\}$ with $|S_{k+1}| = L$ and $\nabla C_A$ is the gradient of $C_A(\mathbf{x})$. Here, $\mathcal{H}_L(\mathbf{x})$ selects the $L$ largest elements of $\mathbf{x}$. In the second stage, the amplitudes at the estimated support are refined as

$$\mathbf{x}^{k+1}(S_{k+1}) = \mathbf{x}^{k+1}(S_{k+1}) - \alpha_{2,k}\,\mathbf{p}^k(S_{k+1}), \tag{24}$$

where $\mathbf{p}^k(S_{k+1})$ is computed using the gradient and Hessian of $\frac{1}{4N_y}\|\mathbf{y} - |\mathbf{Ax}|^2\|_2^2$ at $\mathbf{x}_k$ (cf. Jian-Feng et al.). In the updates equation 23 and equation 24, $\boldsymbol{\alpha} = [\boldsymbol{\alpha_1}\ \boldsymbol{\alpha_2}]$ are the step sizes.

## 6.2 Unfolding and Learnable/Tunable Parameters

Unrolling this algorithm yields a $K$-layer PR network that estimates $\mathbf{x}$ as $f_{\boldsymbol{\alpha},\beta}(\mathbf{y})$. In the unrolling, we learn $\boldsymbol{\alpha} = [\boldsymbol{\alpha_1}\ \boldsymbol{\alpha_2}] \in \mathbb{R}^{K\times2}$, while gradients and Hessians depend only on $\mathbf{A}$ and $\mathbf{x}_k$. In this unfolding, we learn $\boldsymbol{\alpha}$ while updating the gradients/Hessians using $\mathbf{A}$ and the previous-layer's estimate. Importantly, we set $\beta = L = \theta$.

Unlike NA-LISTA and gain calibration, where $\beta$ had no direct interpretation, here $\beta$ is precisely the sparsity level. Thus, in PT and JT, we set $\beta = L$; in P-TDA, $g_{\boldsymbol{\varphi}}(\theta) = \theta$, with no additional learning. Hence, PT, JT, and P-TDA all assume known sparsity. Next, we consider sparsity estimation for DD-TDA.

## 6.3 Sparsity Estimation for DD-TDA

Estimating sparsity from compressed measurements is challenging, particularly in the presence of noise Lopes (2013); Thiruppathirajan et al. (2022). Existing methods often rely on specially designed sensing matrices, such as Gaussian/Cauchy Lopes (2013) or binary constructions Thiruppathirajan et al. (2022). The PR setting is even more difficult since only magnitude measurements are available.

We adapt the approach of Lopes (2013) to PR. Consider an auxiliary measurement $\mathbf{y}_c = |\mathbf{A}_c\mathbf{x}|^2 + \boldsymbol{\eta}_c$, where $\mathbf{A}_c$ is drawn from a Cauchy distribution, while $\mathbf{A}$ in equation 22 is Gaussian. The sparsity can then be estimated as $\left\lceil \hat{T}_1^2(\mathbf{y}_c)/\hat{T}_2^2(\mathbf{y}) \right\rceil$, where

$$\hat{T}_1(\mathbf{y}_c) = \frac{\mathrm{median}(\sqrt{|\mathbf{y}_c|})}{\gamma_c}, \quad \text{and} \quad \hat{T}_2^2(\mathbf{y}) = \frac{\|\sqrt{\mathbf{y}}\|_2^2}{N_y\,\gamma_g^2}, \tag{25}$$

with $\gamma_c, \gamma_g$ depending on the respective distributions and not requiring precise knowledge in practice.

Building on this, we define a sparsity estimator (SE) as

$$\mathrm{SE}_{[\gamma_c,\gamma_g,b]}(\mathbf{y}, \mathbf{y}_c) = \left\lceil \frac{\hat{T}_1^2(\mathbf{y}_c)}{\hat{T}_2^2(\mathbf{y})} + b \right\rceil = N_y\left\lceil \frac{(\mathrm{median}(\sqrt{|\mathbf{y}_c|}))^2\gamma_g^2}{\|\sqrt{\mathbf{y}}\|_2^2\,\gamma_c^2} + b \right\rceil, \tag{26}$$

where the bias term $b$ compensates for estimation errors. The parameters $[\gamma_c, \gamma_g]$, related to the data distribution, are estimated from $[\mathbf{y}, \mathbf{y}_c]$ using a neural network $\Gamma_\phi(\mathbf{y}, \mathbf{y}_c)$ with trainable parameters $\phi$. Hence, the tunable parameter estimator for this problem is given by $h_\varphi(\mathbf{y}, \mathbf{y}_c) = \mathrm{SE}_{[\Gamma_\phi(\mathbf{y},\mathbf{y}_c),b]}(\mathbf{y}, \mathbf{y}_c)$, where the learnable parameters are $\varphi = [\phi, b]$.

Compared to PT, JT, and P-TDA, the DD-TDA method requires $N_c$ additional measurements due to estimator limitations. A better sparsity estimator could avoid the extra measurements. Further, unlike in earlier applications where $\beta$ lacked a direct physical interpretation, here, $\beta$ corresponds to the sparsity level and thus has ground-truth during training. As a result, $h_\varphi$ can be trained independently of the PR module.

The architectures for the different methods for the PR problem are shown in Fig 12.

## 6.4 Results and Discussion

To evaluate the performance of the proposed adaptive phase retrieval framework, we conduct a series of experiments comparing the P-TDA and DD-TDA models against the PT and JT baselines. The domain parameter is the sparsity level $L$ of the signal $\mathbf{x}$.

**Data Generation and Network Architecture:** We generated a synthetic dataset according to the measurement model in equation 22. The measurement matrix $\mathbf{A} \in \mathbb{R}^{N_y \times N_x}$ was constructed with entries drawn from a standard Gaussian distribution, i.e., $\mathbf{A}_{:j} \sim \mathcal{N}(0, \gamma\mathbf{I}_{N_y})$. For the DD-TDA model, an additional measurement matrix $\mathbf{A}_c \in \mathbb{R}^{N_{yc} \times N_x}$ was used for sparsity estimation, with its entries drawn from a standard Cauchy distribution. The ground truth sparse signal $\mathbf{x} \in \mathbb{R}^{N_x}$ was binary, with its non-zero entries randomly set to either $+1$ or $-1$.

The experiments were conducted across three distinct domains defined by their sparsity levels: D1 ($L = 10$), D2 ($L = 7$), and D3 ($L = 4$). We set the signal and measurement dimensions as $N_x = 1700$, $N_y = 1200$, and $N_{yc} = 400$. A total of $N = 6000$ samples were generated, which were then split into 80% training, 15% validation, and the remaining 5% as testing sets.

For the PT approach, three separate models (PT D1, PT D2, PT D3) were trained exclusively on data corresponding to their respective sparsity domains. In contrast, the JT, P-TDA, and DD-TDA models were all trained on a mixed dataset containing samples from all three domains. The unfolded PR network for all models consists of $K = 10$ layers, with the step sizes $\boldsymbol{\alpha}_1$ and $\boldsymbol{\alpha}_2$ as learnable parameters. The key tunable parameter is the sparsity level $\beta = L$ used in the hard-thresholding operator $\mathcal{H}_L(\cdot)$ within each layer. For P-TDA, this parameter is set to the known ground-truth sparsity for each sample.

For DD-TDA, the sparsity is estimated by the network $\mathrm{SE}_{[\Gamma_\phi(.),b]}(.)$ where the function $\Gamma_\phi$ comprises two layers of multi-layer perceptrons. This network is learned first in a separate training stage. Subsequently, these parameters are frozen, and the PR network is integrated into the DD-TDA model, where only $\boldsymbol{\alpha} = [\boldsymbol{\alpha}_1, \boldsymbol{\alpha}_2]$ is trained.

**Performance Metrics:** Model performance was evaluated using two primary metrics, Relative Error (RE) and HR. Given the global phase ambiguity inherent in phase retrieval, the RE between a ground truth signal $\mathbf{x}^*$ and an estimate $\hat{\mathbf{x}}$ is defined to account for potential sign flips as,

$$\mathrm{RE}(\hat{\mathbf{x}}, \mathbf{x}^*) = \min\left(\frac{\|\hat{\mathbf{x}} - \mathbf{x}^*\|_2}{\|\mathbf{x}^*\|_2}, \frac{\|\hat{\mathbf{x}} + \mathbf{x}^*\|_2}{\|\mathbf{x}^*\|_2}\right). \tag{27}$$

The RE is reported in decibels (dB). The HR measures the fraction of correctly identified non-zero elements in the support of $\mathbf{x}^*$ whose estimated amplitudes are within a specified tolerance. For our experiments, the tolerance for HR was set to $t = 0.05$.

**Comparison of Methods:** The performance of the different models on the three test domains is presented in Fig. 13. The results clearly demonstrate the limitations of fixed-parameter models and the advantages of our proposed adaptive approaches.

The PT models, designed as domain specialists, perform well only when the test domain matches their training domain. For instance, PT D3, trained with $L = 4$, achieves a low RE on D3 test data but fails

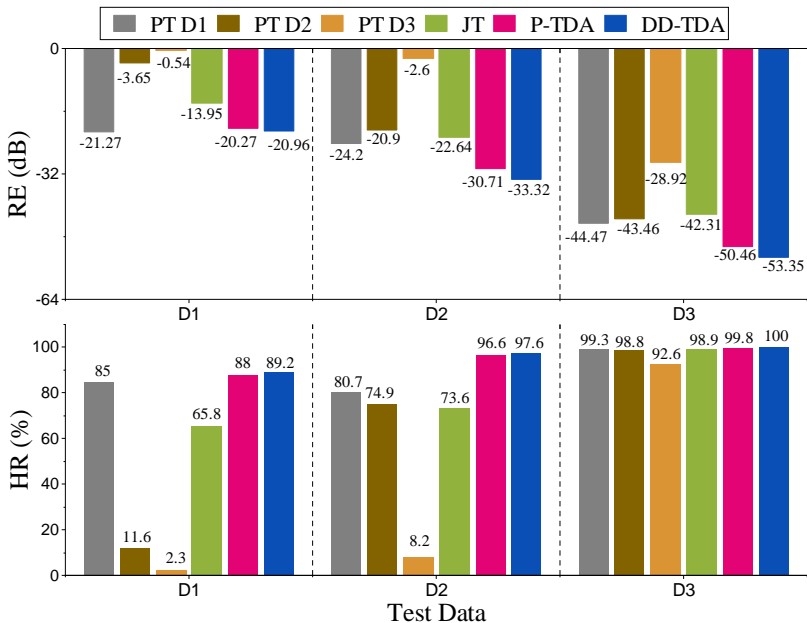

Figure 13: Comparison of the proposed methods with the JT and PT methods for Phase Retrieval with sparsity [10, 7, 4] and HR tol: 0.05.

catastrophically on D1 ($L = 10$) and D2 ($L = 7$). This is because its $\mathcal{H}_L(\cdot)$ operator is fixed to select only four components, making it impossible to recover signals with higher sparsity. Conversely, PT D1 is trained with $L = 10$. When tested on D2 and D3, its operator selects the 10 largest components, which are sufficient to contain the true supports of size seven and four, respectively. This leads to surprisingly robust performance across all domains, although its performance on D1 is not as strong as the adaptive models, indicating that simply over-fixing the sparsity parameter is a suboptimal strategy. The JT model, trained on a mix of all domains by setting $L = 10$, finds a middle ground and delivers average performance across all the domains, failing to match the PT models in their respective domains.

In sharp contrast, both the P-TDA and DD-TDA models exhibit consistently superior performance across all test domains, achieving the lowest RE and highest HR values. This accomplishment is directly related to their capacity to dynamically adjust the sparsity parameter L on a per-sample basis. The P-TDA model, which uses the ground-truth sparsity, sets a high-performance benchmark. The DD-TDA model estimates the sparsity directly from the measurements, performing on par with, and in some cases even slightly better than, the P-TDA model. This performance advantage is due to the learned bias term $b$ in the sparsity estimation network $h_\varphi(\cdot)$. This bias allows the model to learn to slightly overestimate the sparsity, a strategy which can be beneficial in noisy conditions by preventing the premature discarding of true signal components.

Overall, the results confirm that adapting the sparsity parameter is crucial for robust phase retrieval across varying domains. The proposed DD-TDA framework provides an effective, data-driven mechanism to achieve this adaptation, yielding a single model that outperforms generalist baselines.

## 7    Network Complexity and Performance Comparisons

In all the above problems, additional networks $g_\varphi$ and $h_\varphi$ are added to P-TDA and DD-TDA methods, respectively. This results in additional parameters and computations in P-TDA and DD-TDA in comparison to the PT and JT methods, as compared in Fig. 14. The plots show that the P-TDA/DD-TDA methods require considerably higher numbers of parameters and FLOPS. However, we would like to mention that the networks $g_\varphi$ and $h_\varphi$ were intentionally implemented in a non-optimized form, as the focus of this work was to demonstrate the feasibility of tunable adaptation. Their parameter footprint can be greatly reduced in

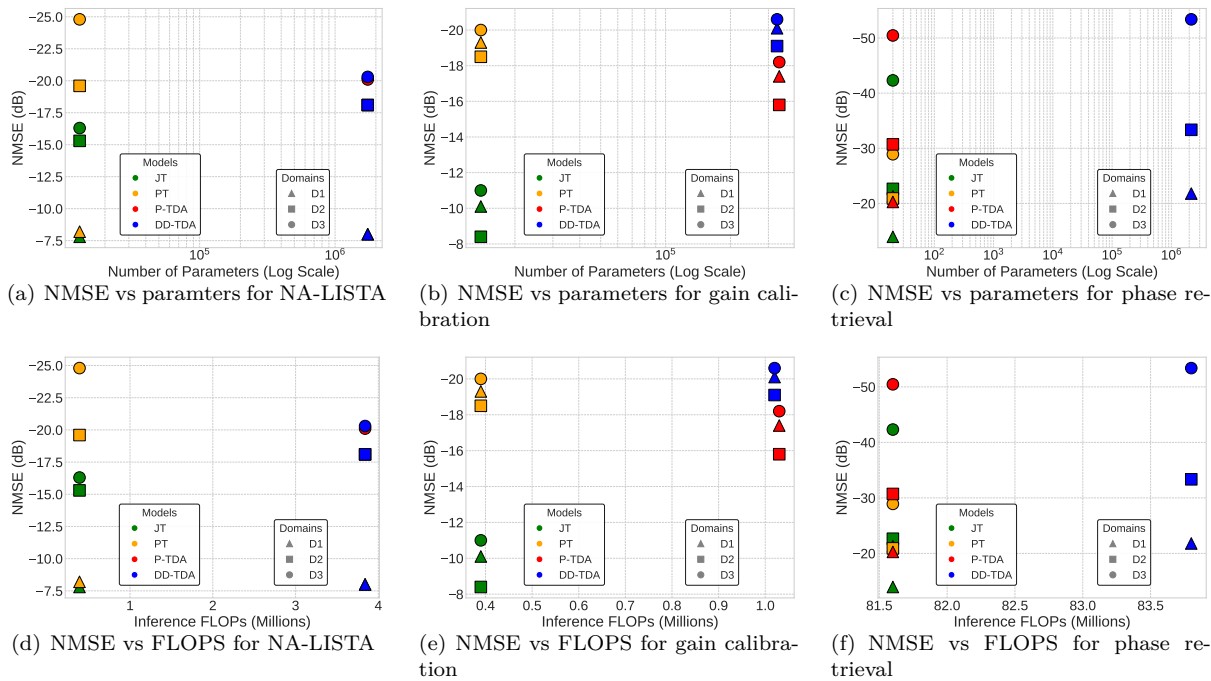

Figure 14: A comparison of NMSE against the number of parameters and FLOPS for different methods.

practice without modifying the underlying methodology. Further, note that PT requires training and storing a separate LISTA model for each domain. Thus, the parameter/FLOP count grows linearly with the number of domains. In contrast, DD-TDA uses one model trained on mixed data and can generalize to new domains without access to the domain labels or parameters.

It is also worth noting from Figs. 14(c) and 14(f) that for the phase retrieval problem, the P-TDA model exhibits a parameter count comparable to the PT and JT methods. However, there is a significant increase in the number of parameters for the DD-TDA model. This jump is because of the architecture of the tuning function, $\Gamma(\cdot)$. In our implementation, $\Gamma(\cdot)$ is a two-layer network that maps a high-dimensional input of size 1600 to an output of size $10 \times 2$. The high input dimension of this network is the main reason for the large parameter footprint. This particular component offers a clear opportunity for optimization, and its architecture could be redesigned to create a more lightweight network, thereby reducing the overall complexity of the DD-TDA model.

## 8  Conclusion

In this paper, we propose a systematic framework for interpretable, unrolled architecture-based domain-adaptive sparse signal recovery with two tunability strategies: P-TDA, which uses known domain parameters, and DD-TDA, which derives domain-relevant variations from the data itself. We concentrate on three realistic and different types of domain variability: additive noise variance, sensor calibration gains, and sparse phase retrieval across domains.

In the case of noise-varying domains, our approaches enable the sparse recovery network to adjust its thresholding behavior without requiring domain-specific retraining. We show that P-TDA can successfully normalize the input using known gains in the more difficult blind calibration scenario. When measurements are affected by unknown gain vectors, DD-TDA learns to correct for gain distortions without explicit access to the calibration parameters. We further demonstrated the framework's versatility on the challenging problem of sparse phase retrieval, where our DD-TDA model successfully adapted to varying signal sparsity levels using a data-driven estimator. Across all applications, the suggested approaches consistently exceeded normal

JT models and closely resembled or matched the performance of domain-specific PT models. In the future, unsupervised domain learning will be used to extend this framework to incorporate more complex models.

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
