# OpenReview forum: "Tunable Domain Adaptation Using Unfolding"
_TMLR — Rejected by TMLR_

### Review · Reviewer_ueXD · 2025-07-16

**Summary Of Contributions:**

This work provides a tunable strategy for LISTA to improve generalization to different noise-levels and gain factor in measurement matrix for compressed sensing task.

**Audience:**

Yes

**Claims And Evidence:**

No

**Requested Changes:**

Please see comments above.

**Strengths And Weaknesses:**

## Strength:

Using tunable parameter to improve generalization to various conditions in regression tasks is an interesting and practical topic.

## Weakness and comments:

I have some major concerns regarding the work and would appreciate it if the authors could address the following questions.

1. The organization and overall writing:

 - (a) missing thoroughly literature in many fields:
     - Works that estimate noise-levels and other distribution-shift related parameters.
     - Methods for solving inverse problems in an adaptive manner, especially those that treat parameters like noise level as control variables.
     - Broader transfer learning and domain adaptation techniques beyond layer-wise retraining; many alternative approaches exist especially for regression problems, and may align with the goals of this work but are not discussed.
     - Figure 2 and 6 does not provide useful information, could omit to allow more space.

- (b) the overall structure is not easy to follow, many redundancies in the paper. The discussions of related work, proposed methods, and experimental results are interwoven throughout the text.

- (c) figures 2 and 6 provide limited information and could be omitted to make space for more interesting experiments.



2. The contributions of the paper are unclear and appear to be overstated:
- (a) the paper claims to propose an adaptive method for general regression tasks, yet it focuses solely on compressed sensing, both in terms of modeling and experimental validation. The assumed model, y = Ax + noise, characterizes an inverse problem rather than a general regression setting. Given the focus on compressed sensing, the paper's conclusions about general regression tasks are not well justified.

- (b) LISTA is one specific unrolling method tailored to compressed sensing. However, there are more general unrolled optimization frameworks for inverse problems that are not discussed. Traditional unrolling approaches typically assume a known forward model A, in which case the learned components (e.g., regularization updates) are not clearly disentangled from parameters such as alpha (condition-independent) and beta (tunable). For scenarios where A is unknown, there exists a rich body of literature that explores adaptive solutions, many of which could provide valuable context but are absent from the discussion.

- (c) the paper mentioned two types of distribution shifts in CS task: noise level and gain in the measurement matrix. Other more realistic distribution-shift settings are not discussed. For example, it is common in CS to have unknown and varying sparsity level K. How does it extend to broader class of regression problems.

- (d) The proposed P-TDA model assumes a known noise level sigma, yet the paper itself acknowledges that sigma is typically unknown in practice, which limits the practical applicability of P-TDA. Furthermore, when sigma is known, Equation (15) allows for a direct estimation of beta without relying on neural network-based learning or cross-validation methods, or beta can also be estimated using cross-validation, as mentioned in the two paragraphs following Figure 2. However, the paper does not provide any empirical comparisons with these simpler alternatives.


3. Experiments
- (a) The experimental evaluation is limited to synthetic datasets. To strengthen the claims, I encourage the authors to consider more realistic scenarios, in particular various more complex generalization scenarios, even within the general scope of inverse problems. In addition, baselines only consider LISTA (in 2010) trained using JT and PT and, I am sure there are other existing works that can serve as baseline models. I recommend also include works that estimate noise levels or other tunable parameters.

- (b) the two subplots in figure 3 are confusing. I understand they correspond to sub-datasets with different noise levels, but splitting six noise levels across two separate plots does not provide clear additional information. The use of labels such as D1, D2, and D3 across both ranges of datasets is ambiguous. Figure 3(b) also includes D2, D4, and D6 in its legend, but not discussed in text. Please clarify and standardize all notations for consistency and readability.

- (c) the comment about Figure 3 says "We observe that PT methods trained for a specific domain perform well when tested on the same domain", but it does not appear true for PT methods in measure of HR. Please revisit and verify this claim.

- (d) I am expecting for better performance using P-TDA than DD-TDA, since P-TDA uses known parameters (noise-level and gain vector), but sometimes it is not, could authors elaborate on this?

- (e) providing visualizations of the reconstructed sparse signal can help interpret the results more straightforwardly.

4. Other technical questions
- (a) the paper mentioned LISTA is shown to converge in fewer iterations, but since it has a fixed number of iterations, it does not necessarily converge.

- (b) in Figure 1, LISTA does not assume known A and update W1 and W2 from y. P-TDA and DD-TDA both use A to estimate beta. However, if A is known, W1 and W2 can be directly derived. It is unclear how this is handled in the experiments. If LISTA is trained without knowledge of A while the proposed methods use it explicitly, this raises fairness concerns in the comparison. Please clarify how these scenarios are aligned and whether additional information is available to P-TDA/DD-TDA that is not provided to LISTA.

- (c) the paper claims that the neural network learn to estimate beta, but it is not clear how beta is compared numerically to ground-truth (if there is one) and other methods such as cross-validation. If the estimated beta has no physical meaning, it may function only as an internal latent variable in a black-box manner. Numerical evaluation of beta would be helpful.

---

> ### Author Response · Authors · 2025-07-29
> **Clarifications and suggested revision plan**
>
> We thank the reviewer for his insightful comments and suggestions. Here are the pointwise responses.
>
> **1, 2b.** We will add literature as suggested. Please let us know if you have something specific. We shall structure the paper to improve clarity and remove Figs 2 and 6.
>
> **2a, c.** We thank the reviewer for the suggestion. While our current experiments focus on structured linear
> inverse problems (sparse recovery and blind gain calibration using LISTA), we agree that broader validation is valuable.
> To address this, we have added a new experiment on phase retrieval, a nonlinear inverse problem
> where the domain shift arises from varying sparsity levels. We build on the unrolled architecture of Naimipour et al. (arXiv:2012.11102), which is independent of LISTA. This addition demonstrates: (i) generalization of our framework to nonlinear problems, (ii) compatibility with non-LISTA architectures, and (iii) handling of domain shifts beyond noise and gain. These results and clarifications will be included in the revised manuscript.
>
> **2d.** We thank the reviewer for raising this important point. We agree that assuming a known noise level $\sigma$ limits the practical use of P-TDA. In our work, P-TDA serves as a proof-of-concept benchmark—an oracle setting illustrating the best-case performance when $\sigma$ is known and mapped analytically to $\beta$.
> While cross-validation could in principle estimate $\beta$, it is impractical in our setting due to (i) the lack of a well-defined candidate set, and (ii) the need for repeated tuning across domains, which hinders scalability under continuous or high-dimensional shifts. In contrast, DD-TDA learns a data-driven mapping to $\beta$ without requiring access to $\sigma$ or per-domain tuning, making it more suitable for adaptive scenarios with latent or dynamic domain factors. We will clarify these distinctions in the revised manuscript.
>
> **3a.** We focused on synthetic datasets to enable controlled studies of domain shifts in noise and gain, allowing us to isolate the effects of model structure and adaptation.
> To address the concern about the scope, we now include a phase retrieval experiment in the revision as discussed.
> We chose LISTA as a canonical backbone for clarity and interpretability. While methods like cross-validation can estimate noise or regularization parameters, they are less scalable under dynamic or high-dimensional domain shifts. In contrast, DD-TDA learns such dependencies directly from data without manual tuning.
>
> **3b.** We shall revise the figure for clarity.
>
> **3c.** Upon revisiting Figure 3, we agree that the claim regarding PT methods performing well on their respective training domains—specifically in terms of HR—was not consistently supported across all domains. We revise our statement accordingly.
>
> **3d.** The fact that DD-TDA occasionally matches or slightly outperforms P-TDA can be attributed to several factors.
>
>  *  DD-TDA learns $\beta$ end-to-end in a task-specific manner, which may lead to more optimal choices under complex domain interactions.
>
> * P-TDA’s fixed $\beta$ to $\sigma$ mapping may be suboptimal when noise interacts with signal structure in ways not captured by the analytical formula.
>
> * DD-TDA may benefit from implicit regularization learned from data, improving robustness to subtle shifts.
>
> * Minor performance gains by DD-TDA may also stem from stochastic effects during training.
>
> **3e.** We add them in the revision.
>
> **4a.**  We will revise the text to clarify that LISTA achieves lower recovery error in fewer layers—compared to ISTA—not that it converges in the traditional optimization sense.
>
> **4b.** In our experiments, all methods—PT, JT, P-TDA, and DD-TDA—assume that the sensing matrix $\mathbf{A}$ is known and fixed, ensuring a fair comparison. In all the methods, $\mathbf{W}_1$ and $\mathbf{W}_2$ are initialized using $\mathbf{A}$.
>
> We also emphasize that if $\mathbf{W}_1$ and $\mathbf{W}_2$ are fixed using ISTA formulas, the accuracy degrades for shallow networks, as ISTA converges slowly and needs many iterations. In contrast, learning these weights improves performance significantly, especially in noisy conditions. This was verified through simulations.
>
> **4c.** We clarify that in DD-TDA, the learned $\beta$ values are interpretable in the sense that they control the thresholding behavior of the unrolled LISTA network, and they vary smoothly and consistently across domains (e.g., as noise level increases). However, we agree that the estimated $\beta$ does not always admit a unique "ground truth" against which it can be directly compared numerically.
>
> We view this deviation not as a drawback, but as an indicator that the learned $\beta$ can implicitly adapt to more complex interactions between data and domain shifts (e.g., sparsity structure, gain perturbations) that are not captured in the closed-form expression.

---

> > ### Author Response · Authors · 2025-08-27
> > **A revision is updated**
> >
> > We have revised the manuscript to address the reviewers’ comments by incorporating the suggested literature, adding clarifications throughout the text, averaging results over multiple runs, and providing new plots to illustrate parameter variance/invariance across domains. We also streamlined the flow to remove redundancies and corrected claims and figure labels in the simulation sections. Additionally, we are working on the requested phase retrieval experiments and will include these results in the next update.

---

> > > ### Author Response · Authors · 2025-09-07
> > > **Revision updated**
> > >
> > > We have revised the manuscript to address the reviewers’ comments and have added a new section on sparse phase retrieval, demonstrating the applicability of the proposed tunability framework to a non-linear inverse problem. We greatly appreciate the reviewers’ insights and constructive feedback, which have helped us improve the clarity and scope of the paper.

---

### Review · Reviewer_34rQ · 2025-07-18

**Summary Of Contributions:**

This paper focuse on resolving the problem of domain adaptation issue in ML models. In ML data, there would be domain shift from cenario to scenario. Training either one model for all domain results in bad transfrability while training multiple models for each different situation is also ineffective. The paper address this question by proposing two data adaptation methods, namely, parametric-tunable domain adaptation (P-TDA) and Data-Driven TDA (DD-TDA). P-TDA uses known information on the domain for TDA while DD-TDA uses the data to infer the domain change. The paper investigates the performance of the proposed method on two problems, including Noise-adaptive LISTA and Domain adaptive Gain Calibration problems. The proposed methods demonstrate comparable performance to domain-specific trained model and outperforms joint trained model.

**Audience:**

Yes

**Claims And Evidence:**

Yes

**Requested Changes:**

Minor points:
In eq(2), $\alpha, \beta$ should have subscript $j$.

**Strengths And Weaknesses:**

Strength:
Clarity:
1. The problem formulation is clearly stated
2. The paper is well-motivated.
3. The data and model construction are clearly stated.
Novelty:
1. The proposed work demonstrates good performance compared with the baseline methods including PT and JT.

Weakness:
Significance:
1. From the example problems provided, i.e., Noise-adaptive LISTA and Domain adaptive Gain Calibration, it seems like the model requires to be interpretable. Suppose, for example, in a more complex problem, such as image classification, where we don't know the shape of $\beta$ that decides the data distribution, how can we decide where and how we place $h_{\phi}$ or $g_{\phi}$ in the original network? I think this might undermines the significane of the proposed work.
2. The problem setting is fairly simple. The unlderlying data structure in the experiments are only in linear relationship. It is good if the paper could add some experiments on nonlinear problems (e.g., phase retrieval problem?)

Novelty:
1. The paper failed to compare the proposed method with Domain-Adversarial training method, e.g., "Domain-Adversarial Training of Neural Networks, JMLR 2016", which also introduces an extra subnetwork for domain-related parameter prediction.

---

> ### Author Response · Authors · 2025-07-29
> **Clarifications and suggested revision plan**
>
> We appreciate your helpful review. Your comments greatly enhanced our manuscript. Our responses are below. We believe these responses are satisfactory for the revised version and hope the manuscript is now ready for publication.
>
> **Strength: Clarity:**
>
> **Response:** We thank the reviewer for their positive feedback and thoughtful evaluation of our work.
> We are glad to hear that the problem formulation, motivation, and methodology were found to be clear. We also appreciate the reviewer’s recognition of the performance improvement over baseline methods (JT and PT), and the novelty of our proposed approach. We will continue to improve the manuscript's clarity and organization in response to specific suggestions to further enhance its quality.
>
> **Weakness: Significance:**
>
> 1.  **Response:** We thank the reviewer for raising this insightful concern. We interpret the reviewer’s reference to $\beta$ as intending to denote a domain descriptor, which we denote as $\theta$—a latent or explicit parameter that governs the distribution shift (e.g., noise level, gain, blur strength, etc.). In our framework, $\beta$ is a tunable model parameter whose value is adapted based on $\theta$.
>     We agree that our framework is best suited to tasks where such domain parameters $\theta$ are known, structured, or interpretable. This includes a broad class of inverse problems and structured regression tasks, where the location and role of $\theta$ within the model (e.g., as a threshold parameter, step size, or regularizer) are well-defined and motivate the placement of tunable modules like $h_\phi$ or $g_\phi$.
>     In contrast, in tasks such as image classification, the nature of domain shifts is often high-dimensional and unstructured, making it less clear how to parameterize $\theta$, or where to inject a domain-adaptive block. We fully agree that extending our approach to such settings would require either (i) unsupervised discovery of latent domain parameters or (ii) integration with black-box domain generalization techniques. These are important directions for future work but fall outside the structured and interpretable setting we focus on in this paper.
>     We will revise the manuscript to clarify these assumptions and to more precisely distinguish between $\theta$ (domain parameter) and $\beta$ (model parameter) in the context of adaptation.
>
> 2.  **Response:** We thank the reviewer for this helpful suggestion. We agree that evaluating the proposed framework on nonlinear inverse problems would strengthen the empirical validation and demonstrate broader applicability.
>     In response, we have included an additional experiment on the **phase retrieval problem** as suggested, a canonical nonlinear inverse problem where the goal is to recover a sparse signal from the magnitude of linear measurements. We build upon the unfolding strategy proposed by Naimipour et al. (“Unfolded Algorithms for Deep Phase Retrieval”, arXiv:2012.11102) and incorporate domain shifts in the sparsity level, which serves as the tunable parameter.
>     This experiment highlights several key points:
>     * The proposed domain-adaptive tuning strategy generalizes beyond linear problems.
>     * It applies to unrolled architectures beyond LISTA.
>     * It is effective in handling domain shifts that are not limited to noise or gain, such as variation in signal sparsity.
>
>     We will describe this new result in detail in the revised manuscript and include performance comparisons between PT, JT, and our proposed DD-TDA strategy in this nonlinear setting.
>
> **Novelty:**
>
> 1.  **Response:** While Domain-Adversarial Training of Neural Networks (Ganin et al., JMLR 2016) indeed introduces an auxiliary subnetwork to encourage domain invariance and modulate domain-specific adaptation, the approach differs significantly in architectural and conceptual terms. That method operates within a standard feed-forward classification network, with generic parameters in the earlier layers and domain-adaptive features learned in later layers.
>     In contrast, our approach is based on unrolled optimization networks, which offer a layer-wise interpretability that traditional feed-forward architectures lack. This interpretability allows us to explicitly identify and adapt domain-specific parameters across the iterations of the unfolded solver, enabling finer control over the adaptation process in inverse problems. Thus, while both approaches aim to address domain shift, the underlying mechanisms and applications are fundamentally different.
>
> **Requested Changes:**
>
> **Response:** In Equation (2), $\boldsymbol{\alpha, \beta}$ are the dummy variables, and the output of the optimization algorithm would be $\boldsymbol{\alpha_j, \beta_j}$. We could change them as suggested.

---

> > ### Author Response · Authors · 2025-08-27
> > **A revision is uploaded**
> >
> > We have revised the manuscript to address the reviewers’ comments by incorporating the suggested literature, adding clarifications throughout the text, averaging results over multiple runs, and providing new plots to illustrate parameter variance/invariance across domains. We also streamlined the flow to remove redundancies and corrected claims and figure labels in the simulation sections. Additionally, we are working on the requested phase retrieval experiments and will include these results in the next update.

---

> > > ### Author Response · Authors · 2025-09-07
> > > **Revision updated**
> > >
> > > We have revised the manuscript to address the reviewers’ comments and have added a new section on sparse phase retrieval, demonstrating the applicability of the proposed tunability framework to a non-linear inverse problem. We greatly appreciate the reviewers’ insights and constructive feedback, which have helped us improve the clarity and scope of the paper.

---

### Review · Reviewer_opZn · 2025-08-16

**Summary Of Contributions:**

This paper presents a novel set of methods for domain adaptation for regression tasks. Specifically, this work targets the case in which data from different domains of a task (domains differing for example in the level of additive or multiplicative noise at the sensor/input stage) may be processed by a single type of model which can leverage parametric or data-driven estimation of tunable model parameters to adapt between domains. The efficacy of this method is shown on a simple set of regression tasks with additive noise or multiplicative (sensor-gain change) noise. This work neatly lays out multiple formulations for solutions to such a problem and shows how it is possible to build tunable parameters into interpretable unrolled networks. It is shown that the novel proposed methods can match or exceed the performance of the commonly used alternatives, namely the training of personalized models for each domain or a joint model across all domains.

**Audience:**

Yes

**Claims And Evidence:**

Yes

**Requested Changes:**

The paper is a neat and nice contribution, with perhaps limited strength. However, for a few reasons I would lean to rejection at the moment. If the points of change I have requested below can be addressed, I would be happy to revise my sense of this paper's rating.

__Issues requiring attention:__
- *Significance and error bars*: The results shown in all figures show only a single run result for each algorithm under each condition. These results are limited enough (and in some cases close enough to PT/JT) that I am unconvinced that the results are significant. Running all results for five repeats seems like a computationally achievable change which would allow for error bars and/or significance testing so that the true order of method efficacies can be revealed.
- *Lack of clarity around $\alpha$ vs $\beta$ changes*: In the text (p10), a metric ($S(\cdot)$) is described for measuring the size of parameters changes. Based upon Table 2, it is claimed that there is evidence for a significant change in $\beta$ and non-significant change in the metrics measured. However this does not appear to be the case (across domains the metric changes a great deal). More importantly, the claim being made here and the evidence are not compatible at worse and at best are unclear. Adding more clarity would help a reader to appreciate the point better, and an alternative metric may be necessary to support the current conclusion.
- *P-TDA vs DD-TDA*: In the abstract and introduction, Parametric vs Data-Driven distinctions are made between the two core proposed variants of the tunable domain adaptive algorithms. However, on closer investigation of the methods it is clear that the only distinction between these two methods is the inclusion of the domain defining parameter as an input to ($g(\cdot)$). Though it is fair to describe this as inclusive of a parameter, nothing in principle stops the P-TDA method to make use of "Data-Driven" information along with this parameter. Making this more clear in the overall claims of this work would make it more honest. At minimum this should be discussed in the results and conclusions.
- *Computational/Memory Complexity*: It would be much appreciated if the number of parameters required for each method and the FLOPS (computational complexity) are made more clear. It is currently unclear whether the extra network computations and parameters required by P-TDA and DD-TDA are significantly more that the compared PT/JT methods. This is important when considering whether one aught to use multiple PT networks or a DD-TDA network, for example.


__Textual issues:__
- p3: $N_y$ and $N_x$ are not defined
- p3: Equation two should be $\{\alpha_j, \beta_j\} = arg min ...$
- p5: Section 4.1: In the description of the Compressed Sensing Problem, the gaussian noise vector should be described with a vector of means (zero vector) and a diagonal covariance matrix ($\sigma^2 I$) and can be done so as is done for equation 13. Currently it is described as if it were a univariate gaussian.
- p6: Section 4.2: Text repetition: "Aberdam et al. Aberdam et al. (2021) introduced "
- Figure 1: b and c should have more explanatory text in the caption
- p8: "Data Generation and Network Architecture" section needs some corrections. There is a dot after P-TDA in the first sentence of the second paragraph which shouldn't be there. And an excess comma later in the same sentence.
- Figure 4: More honesty would be appreciated in the caption since the models proposed are in fact rarely better than PT. Furthemore, the legend indicates "PT D2", "PT D4", and "PT D6" when the text suggests that these are in fact "PT D1", "PT D3", and "PT D5". Either clarify the text or update the caption

**Strengths And Weaknesses:**

__Strengths__
- Well written/structured methods section describing the core problem and proposed solutions
- Neat and simple approach with well evidenced applicability
- A good/integrated contribution of introduction, theory, and practical experiments

__Weaknesses__
- Margins of outperformance are small and not significance tested
- The claim that the tunable parameter is the only affected parameter by tuning is weakly supported
- The two methods P-TDA and DD-TDA are less different in detail than claimed
- It is unclear what the computational or parameter differences are between the compared methods
- Some textual issues could be resolved

---

> ### Author Response · Authors · 2025-08-18
> **Clarifications and suggested revision plan**
>
> ## Response to Reviewer opZn
>
>
> We appreciate the reviewer’s careful reading and constructive suggestions. Below we address each requested change and textual issue. Revisions have been (or are being) incorporated accordingly.
>
> ### 1. Significance and error bars
> We have begun our simulations. We will report the results as soon as we have them.
>
> ### 2. Clarity on α vs β changes
> We understand the concern. We would like to highlight that the matrices $\mathbf{W}_1$ and $\mathbf{W}_2$ are of size $100 \times 30$ and $100 \times 100$, respectively. The variation in their measures is less than two. Whereas, the change in $\beta$ is significant compared to the true value. To clarify this further, we computed the relative changes in $S(\mathbf{W}_1)$, $S(\mathbf{W}_2)$, and $\beta$ for the second and third domains, with respect to the first one.
>
>
> * The relative changes in  $S(\mathbf{W}_1)$ s are $1.5$ % and $4.9$ %, respectively for D2 and D3.
>
> * The relative changes in  $S(\mathbf{W}_2)$ s are $18$ % and $3.6$ %, respectively for D2 and D3.
>
> * Whereas, the changes in $\beta$ s are 45.5% and 58%, respectively. The metric shows that $\beta$ varies significantly than the $\alpha$.
>
> ### 3. P-TDA vs DD-TDA distinction
> Thank you for this observation. We agree and would like to emphasize (as also mentioned in a previous response) that P-TDA is not proposed as a practical method, but rather serves as a proof-of-concept upper bound that illustrates the performance one could achieve if the domain parameter were known and used directly for tuning. In contrast, DD-TDA reflects the realistic case where the domain parameter is not available and must be inferred from the data. We will clarify this more explicitly in the abstract and conclusion to avoid suggesting that P-TDA and DD-TDA are two fundamentally different “solution classes” — rather, P-TDA is used as an oracle baseline to validate our design and motivate the data-driven version (DD-TDA).
>
> ### 4. Computational / memory complexity
>
> For both experiments (noise tunability and gain calibration), the memory and computational requirements are as follows:
>
> - **PT/JT (per domain):** $\approx$ 13,000 parameters; $\approx$ 0.39M FLOPs
> - **P-TDA / DD-TDA (single model handling all domains):** $\approx$ 1,741,513 parameters; $\approx$ 3.84M FLOPs
>
> Note that PT requires training and storing a *separate* LISTA model for each domain. Thus, the parameter/FLOP count grows linearly with the number of domains (e.g., three PT models would already exceed the parameter budget of a single P-TDA/DD-TDA model). In contrast, DD-TDA uses *one* model trained on mixed data and can generalize to new domains without access to domain labels or parameters.
>
> *We also note that the auxiliary networks $h_{\phi}$ and $g_{\phi}$ were intentionally implemented in a non-optimized form, as the focus of this work was to demonstrate the feasibility of tunable adaptation. Their parameter footprint can be greatly reduced in practice (e.g., via bottleneck layers or shared weights) without modifying the underlying methodology.*
>
>
> ---
>
> ## Textual / Formatting Corrections
> We thank the reviewers for pointing out the typos/corrections. We shall fix them in the revision.

---

> > ### Comment · Reviewer_opZn · 2025-08-19
> >
> > Thank you for your response. Aside from awaiting the significance/error bars, I'd also like to point out that I appear to have been misunderstood on point 2 and would desire a bit more detail on point 4.
> >
> > ### 2. Clarity on α vs β changes
> > My issue is not that the variation in your measure is big or small. My issue is that your metric seems to say very little and is rather arbitrary (not well motivated). Within it is a division by the maximum value of W and the reasoning as to why this is not simply the Frobenius norm alone is not clear. What is the hypothesis? Why not just measure the variance in the weights? Are there not individual parameters in $W$ which vary as much as $\beta$ does? I'm not asking these questions for answers but to point out that it is unclear why this metric was chosen and what it is in fact showing. More clarity on your goal here and a theoretically motivated metric would be appreciated.
> >
> > ### 4. Computational / memory complexity
> > In your response it seems clear that there is a significant difference in the number of parameters and FLOPs of the PT/JT and P/DD-TDA models. Regardless of their single vs multi-domain setups (which note, JT also handles multiple domains), this is a very important point to make clear in the paper. Ultimately it seems that for a limited number of domains it would be far more efficient to train multiple PT models or a JT model. I would ideally like to see a plot which shows model performances (y-axis) against flops (x-axis) and separately against number of parameters (x-axis). Even if your methods are not highly optimised, it is good to put the current state in to a perspective so that it can be determined how far from practical the methods currently are.

---

> > > ### Author Response · Authors · 2025-08-20
> > > **Response**
> > >
> > > Here are the clarifications and responses.
> > >
> > > **2.** Thank you for the clarification — this is a fair point. Our intention was not to claim that the chosen metric is fundamental, but simply to illustrate that the tunable parameter $\beta$ varies significantly more across domains than the weight matrices~$W_1$ and $W_2$. The division by the maximum entry was used only as a simple normalization (in the same spirit as NMSE), so that the values are on a comparable scale; we realise that this was not clearly stated in the original text.
> > >
> > > To make this more transparent, we will:
> > > * Clearly state that the goal is to compare the *relative variation* of $\beta$ and the weights across domains,
> > > * Include visualizations of the individual entries of $W_1$ and $W_2$ across domains, showing that they remain nearly constant,
> > > * Revise the paragraph on p.10 to reflect this motivation more explicitly.
> > >
> > > We believe these additions will make the intended conclusion clearer to the reader and avoid any ambiguity regarding the metric.
> > >
> > > **4.** Sure. We are preparing the plots and will add them to the revised manuscript. We will share the revision and update you ASAP.

---

> > > > ### Author Response · Authors · 2025-08-27
> > > > **A revision is added**
> > > >
> > > > We have revised the manuscript to address the reviewers’ comments by incorporating the suggested literature, adding clarifications throughout the text, averaging results over multiple runs, and providing new plots to illustrate parameter variance/invariance across domains. We also streamlined the flow to remove redundancies and corrected claims and figure labels in the simulation sections. Additionally, we are working on the requested phase retrieval experiments and will include these results in the next update.

---

> > > > > ### Author Response · Authors · 2025-09-07
> > > > > **Revision updated with phase retrieval results**
> > > > >
> > > > > We have revised the manuscript to address the reviewers’ comments and have added a new section on sparse phase retrieval, demonstrating the applicability of the proposed tunability framework to a nonlinear inverse problem. We greatly appreciate the reviewers’ insights and constructive feedback, which have helped us improve the clarity and scope of the paper.

---

### Decision · Action_Editor_ntkn · 2025-09-28

**Recommendation:** Reject

**Audience:**

Yes

**Audience Explanation:**

Domain adaptation for regression tasks is an important research problem, and some TMLR's audience would be interested in it.

**Claims And Evidence:**

No

**Claims Explanation:**

This paper presents new domain adaptation methods for regression tasks by using interpretable unrolled networks. Reviewers agreed that the paper is well motivated and clearly written, but also raised many concerns regarding problem setting, technical details, experiments, and complexity. The authors have provided detailed responses, which have addressed some of the concerns from reviewers. However, during the discussion phase, reviewers found that the paper still has some major limitations. For example, the claim of contributing to general regression problems is unconvincing, as the experiments are limited to two inverse problems and lack convincing baseline comparisons. In addition, the complexity of the proposed method is also a concern.